# COMPOSE YOUR POLICIES! IMPROVING DIFFUSION-BASED OR FLOW-BASED ROBOT POLICIES VIA TEST-TIME DISTRIBUTION-LEVEL COMPOSITION

**Jiahang Cao**[1,2,3*]   **Yize Huang**[1,4*]   **Hanzhong Guo**[1]   **Rui Zhang**[1]   **Mu Nan**[1]
**Weijian Mai**[1]   **Jiaxu Wang**[5]   **Hao Cheng**[5]   **Jingkai Sun**[1,2]   **Gang Han**[2]
**Wen Zhao**[2]   **Qiang Zhang**[2]   **Yijie Guo**[2]   **Qihao Zheng**[3]   **Chunfeng Song**[3]
**Xiao Li**[4]   **Ping Luo**[1]   **Andrew F. Luo**[1†]
[1]The University of Hong Kong   [2]Beijing Innovation Center of Humanoid Robotics
[3]Shanghai AI Lab   [4]Shanghai Jiaotong University
[5]The Hong Kong University of Science and Technology

## ABSTRACT

Diffusion-based models for robotic control, including vision-language-action (VLA) and vision-action (VA) policies, have demonstrated significant capabilities. Yet their advancement is constrained by the high cost of acquiring large-scale interaction datasets. This work introduces an alternative paradigm for enhancing policy performance *without additional model training*. Perhaps surprisingly, we demonstrate that the composed policies can exceed the performance of either parent policy. Our contribution is threefold. First, we establish a theoretical foundation showing that the convex composition of distributional scores from multiple diffusion models can yield a superior one-step functional objective compared to any individual score. A Grönwall-type bound is then used to show that this single-step improvement propagates through entire generation trajectories, leading to systemic performance gains. Second, motivated by these results, we propose General Policy Composition (GPC), a training-free method that enhances performance by combining the distributional scores of multiple pre-trained policies via a convex combination and test-time search. GPC is versatile, allowing for the plug-and-play composition of heterogeneous policies, including VA and VLA models, as well as those based on diffusion or flow-matching, irrespective of their input visual modalities. Third, we provide extensive empirical validation. Experiments on Robomimic, PushT, and RoboTwin benchmarks, alongside real-world robotic evaluations, confirm that GPC consistently improves performance and adaptability across a diverse set of tasks. Further analysis of alternative composition operators and weighting strategies offers insights into the mechanisms underlying the success of GPC. These results establish GPC as a simple yet effective method for improving control performance by leveraging existing policies. Our project page is in https://sagecao1125.github.io/GPC-Site/.

## 1 INTRODUCTION

Diffusion Policies (DPs) (Chi et al., 2023; Ho et al., 2020; Song et al., 2020a) have emerged as a powerful method for policy parameterization in robot learning, enabling the representation of complex, multi-modal action distributions – a key advantage for policies conditioning on high-dimensional inputs like vision and language in domains from manipulation (Ze et al., 2024b; Zhu et al., 2024; Liu et al., 2024a) to navigation (Sridhar et al., 2024; Zhang et al., 2024a). Despite this progress, the advancement of diffusion- and flow-based policies is fundamentally constrained by scaling challenges related to both model capacity and data availability. Performance can plateau due to the intrinsic representational limits of a given model, yet scaling up the model architecture also requires

---

*Equal contribution `jiahang@connect.hku.hk`
†Corresponding author `aluo@hku.hk`

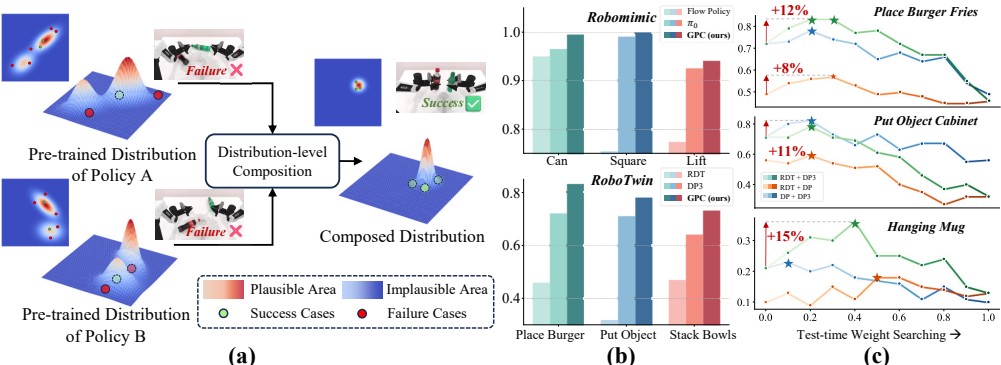

Figure 1: **Illustration of General Policy Composition.** **(a)** Distributions from pre-trained state-of-the-art diffusion- or flow-based policies can be composed to construct a stronger policy without additional training, with a test-time search over composition weights picking the best parent-policy mix; score composition corresponds to the product of probabilistic density functions (PDFs), steering sampling toward consensus regions. **(b)** GPC can yield consistent gains across a diverse set of tasks. **(c)** We find the optimal weight when composing two models can vary depending on the task.

the collection of costly interaction datasets to fully capture the potential performance benefit (Black et al., 2024). Conventional post-training strategies offer limited solutions; supervised fine-tuning requires expensive data collection (Ouyang et al., 2022), while reinforcement learning introduces the complexity of reward engineering and extensive online interaction (Hu et al., 2025).

To overcome these limitations, this work introduces an alternative paradigm: creating stronger policies by composing existing, pre-trained models. While prior work has explored static model composition (Du & Kaelbling, 2024; Wang et al., 2024c), we find that the optimal weighting is not universal but is instead highly task-dependent, even for a fixed set of parent policies. Drawing inspiration from compositional generative modeling, we first establish a theoretical foundation showing that a convex combination of distributional scores can yield a provably superior objective for policy improvement. This principle underpins our proposed method of *General Policy Composition* (GPC, Fig. 1). GPC is a training-free framework that, at inference time, combines the distributional scores of multiple pre-trained policies via convex combination and test-time search. This approach flexibly integrates heterogeneous models – spanning diffusion- and flow-based architectures, VA and VLA modalities, and diverse sensory inputs – to form a more capable policy, all without modifying the base models. Crucially, we demonstrate that *the resulting composed policy can exceed the performance of any of its individual parent policies*.

We validate GPC through extensive experiments in both simulation and real-world environments, demonstrating consistent outperformance against single-policy baselines. Our analysis extends to alternative composition operators (e.g., logical AND/OR) and various weighting configurations, offering broader insights into why and when composition is effective. Our contributions are summarized as follows: **(i)** We establish a theoretical foundation for robot policy composition, proving that the convex combination of distributional scores can yield an improved functional objective and that this advantage propagates to the system level. **(ii)** We propose General Policy Composition (GPC), a flexible, training-free framework that combines pre-trained policies across different modalities and architectures into a more expressive policy. **(iii)** We conduct extensive evaluations in simulation and the real world, demonstrating the consistent performance gains of GPC while analyzing key design choices to guide future research in policy composition.

## 2 RELATED WORK

**Composable Generative Models.** Composability refers to the ability to combine multiple components or distributions into a unified representation while preserving the properties of the individual elements. (i) Visual Generation: Energy-based models (EBMs) (Hinton, 2002; Du & Mordatch, 2019; Grathwohl et al., 2020) support compositionality by summing energies, allowing factor-level combinations. Du et al. (2020) unified perspectives of compositionality for visual generation. Liu et al. (2021) further improved EBMs for scene generation by factorizing relational structures. Skreta et al. (2024) introduced the superposition of diffusion models by using itô estimation. (ii) Language

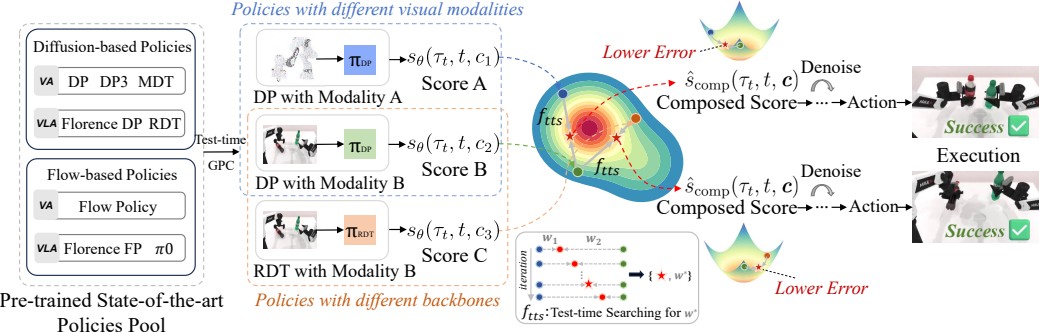

Figure 2: **Overview of our proposed General Policy Composition.** Combining distributional scores from pre-trained diffusion-based or flow-based policies on different conditions (*e.g.*, visual modalities and network backbones), GPC can generate expressive and adaptable action trajectories through convex score combination *without additional training*.

Generation: Du et al. (2023b) combined outputs using multi-agent debate for robust language generation. Lifshitz et al. (2025) proposed multi-agent verification at test-time for improvement.

**Diffusion Models in Robot Learning.** Due to their flexibility and representational power, diffusion models (Ho et al., 2020; Song et al., 2020a; Nichol & Dhariwal, 2021) offer a novel way to represent policies. The concept of Diffusion Policy (Chi et al., 2023) was first proposed to model action spaces using diffusion, significantly enhancing expressiveness. Since then, numerous advancements have been made: multimodal DP such as MDT (Reuss et al., 2024), trajectory extraction approaches like AWE (Shi et al., 2023). DP3 (Ze et al., 2024b) utilizes point cloud representations to achieve state-of-the-art performance, and VLA models, *e.g.*, Octo (Team et al., 2024), $\pi_0$ (Black et al., 2024) and RDT (Liu et al., 2024a). In this work, our GPC can be adopted to various general diffusion-based (*e.g.*, DP, DP3, MDT, and RDT) or flow-based policies (*e.g.*, flow policy and $\pi_0$), demonstrating great flexibility (Detailed in App. G).

**Compositional Diffusion Models in Robotics.** Recent work has explored the use of compositional diffusion models in robotics. Janner et al. (2022) applied compositionality to diffusion-based planning. Yang et al. (2023b) tackled continuous constraint satisfaction in robotic planning, and Luo et al. (2024) improved motion planning by learning potential fields. In addition, Policy Composition (PoCo) (Wang et al., 2024c) does constraint-based, task-based, and input modality-based composition; however, it does not explore the weights between policies. Moreover, it has not been validated in widely adopted simulation environments and provides limited analysis of the underlying composition mechanisms. In contrast, our proposed GPC framework offers broader generality by enabling composition across both VA and VLA models, regardless of input visual modality, and we also deliver deeper insights into the task-dependent weight searching for policy composition.

## 3 PRELIMINARIES

Diffusion models (Sohl-Dickstein et al., 2015) are based on a generative process that iteratively denoises a random noise distribution to generate samples. The following equation describes the update rule for the diffusion process based on Langevin dynamics (Song et al., 2020b):

$$\tau_{t-1} = \alpha_t\,\tau_t + \beta_t\,s_\theta(\tau_t, t) + \gamma_t\,\eta, \quad \eta \sim \mathcal{N}(\mathbf{0}, \sigma_t^2\mathbf{I}), \tag{1}$$

where $s_\theta(\tau_t, t)$ denotes the learned score function, $\alpha_t, \beta_t, \gamma_t$ are coefficients determined by the noise schedule and the choice of solver. Different sampling methods, such as DDPM (Ho et al., 2020), DDIM (Song et al., 2020a), or ODE/SDE-based solvers (Song et al., 2020b), can be recovered by specifying these coefficients accordingly.

Closely related to diffusion models are energy-based models (Hinton, 2002; Du & Mordatch, 2019), which define probability distributions through learnable energy functions. The connection arises because the gradient of the energy function in EBMs plays a role analogous to the score function in diffusion models. Further progress is made with compositional EBMs (Du et al., 2020), where multiple energy functions can be combined by summing their contributions.

## 4 THEORETICAL ANALYSIS OF CONVEX SCORE COMPOSITION

We first provide a mathematical justification for why convex score composition can improve policy performance. Our analysis shows that **(i)** at the functional level, convex combinations of scores from pre-trained policies can yield lower score error, and **(ii)** at the system level, sampling error is bounded by score error through the stability of the sampling dynamics. These results establish convex score composition as a principled foundation for policy improvement, which directly motivates our proposed General Policy Composition framework (in Sec. 5).

### 4.1 FUNCTIONAL-LEVEL IMPROVEMENT

We begin with the question of whether combining score estimators can yield better approximations to the true score $s^*$. The following result shows that there exists a convex combination of two estimators whose error is no greater than that of the better individual estimator, and strictly smaller unless their errors are perfectly aligned.

**Proposition 4.1** (Single-step improvement via convex combination). *Let two score estimators be $\varepsilon_1 = s^* + b_1 + \eta_1$ and $\varepsilon_2 = s^* + b_2 + \eta_2$, with deterministic biases $b_i$ and random zero-mean noise $\eta_i$ that plays the role of the diffusion component in the time-reversed stochastic dynamics (e.g., a reverse-time ODE). For any convex weight $w \in [0,1]$, define $\varepsilon(w) = w\varepsilon_1 + (1-w)\varepsilon_2$. Then the mean-squared error $Q(w) = \mathbb{E}\|\varepsilon(w) - s^*\|^2$ is a convex quadratic in $w$. Its minimizer $w^\star$ satisfies*

$$Q(w^\star) \leqslant \min\{Q(0), Q(1)\},$$

*with strict inequality in most non-trivial cases.*

See proof in App. B. Intuitively, each estimator deviates from the true score in a different way. A convex combination can cancel out these errors, achieving a better score estimator. Unless the two models make identical errors, the true score $s^*$ lies closer to some interior point, ensuring that a weighted average achieves smaller error. This establishes that convex score composition can reduce estimation error at each step.

### 4.2 SYSTEM-LEVEL STABILITY

While Prop. 4.1 shows improvement at the functional level, it remains to understand how score errors propagate into trajectory sampling. The following proposition establishes a stability guarantee: the terminal error is controlled by the cumulative score error.

**Proposition 4.2** (Score-to-sample stability). *Let $x^*(t)$ denote the oracle trajectory derived from the true score $s^*$, and $x_{\hat{s}}(t)$ denote the approximate trajectory derived from an estimator $\hat{s}$, both starting from the same initial condition. They satisfy*

$$\dot{x}^*(t) = F(t, x^*(t), s^*(t, x^*(t))), \quad \dot{x}_{\hat{s}}(t) = F(t, x_{\hat{s}}(t), \hat{s}(t, x_{\hat{s}}(t))),$$

*where $F$ represents the underlying dynamics that map the score into the state update. Suppose $F$ is Lipschitz in $(x, s)$ with constants $L_x(t), L_s(t)$, and $\hat{s}$ is Lipschitz in $x$ with constant $\hat{\Lambda}(t)$. Assume the score error admits a uniform bound $\kappa(t)$ (i.e., $\|\hat{s} - s^*\| \leqslant \kappa(t)$). Define $\tilde{L}(t) = L_x(t) + L_s(t)\hat{\Lambda}(t)$. Then for all $T > 0$,*

$$\mathbb{E}\|x_{\hat{s}}(T) - x^*(T)\| \leqslant \left(\int_0^T e^{2\int_t^T \tilde{L}(\tau)\,d\tau} L_s(t)^2\,dt\right)^{1/2} \left(\int_0^T \kappa(t)^2\,dt\right)^{1/2}.$$

See proof in App. C. This result shows that the sampling dynamics are stable: the terminal error grows at most exponentially with the Lipschitz constants, and is directly bounded by the integrated score error. Thus, reducing score error at each step translates to reducing the overall trajectory error.

### 4.3 IMPLICATIONS FOR POLICY COMPOSITION

Combining Prop. 4.1 and Prop. 4.2 yields a direct implication for composed policies.

> **Corollary 4.1** (Convex score combination tightens the sampling error bound). *Let $\mathcal{B}(\hat{s})$ denote the upper bound on the expected sampling error derived in Proposition 4.2 (specifically Eq. 21). If a convex combination $s_{\text{comp}} = ws_1 + (1-w)s_2$ satisfies*
>
> $$\int_0^T \mathbb{E}\|s_{\text{comp}} - s^*\|^2 dt < \min_i \int_0^T \mathbb{E}\|s_i - s^*\|^2 dt,$$
>
> *then the corresponding theoretical error bound is strictly reduced:*
>
> $$\mathcal{B}(s_{\text{comp}}) < \min_i \mathcal{B}(s_i).$$

See proof in App. D. Once functional-level improvement is established by obtaining an optimal $w^*$ (Prop. 4.1), stability ensures this advantage propagates along the trajectory (Prop. 4.2), making convex score composition provably superior to relying on individual scores.

This theoretical analysis provides a clear justification for convex score composition: it can improve accuracy at each functional step and propagate this advantage through stable sampling dynamics, leading to system-level gains. These results directly motivate GPC, which leverages convex score combination to build stronger policies from pre-trained components. While the theory guarantees the existence of optimal weights, finding them analytically is intractable; hence, in practice we employ test-time searching to identify effective weighting strategies, as explored in Sec. 6.

## 5 OUR METHOD: GENERAL POLICY COMPOSITION

Building on the mathematical foundation in Sec. 4, we now present our method, General Policy Composition, as illustrated in Fig. 2. The key idea is to leverage convex score composition to combine multiple pre-trained policies into a stronger and more expressive one. We first revisit the mathematical formulation of compositional diffusion models in Sec. 5.1, which provides a basis for composing policies conditioned on different factors. We then introduce our method in Sec. 5.2, where GPC convexly combines the scores of diffusion or flow-based policies across modalities, architectures, or VA/VLA settings. Finally, we extend this framework in Sec. 5.3 to include alternative composition operators, offering a broader view of policy composition beyond convex averaging.

### 5.1 COMPOSITIONAL DIFFUSION MODELS

The key idea of the compositional diffusion model (CDM) is to model the distribution of a trajectory $\tau$ conditioned on multiple concepts $c_i$, similar to the compositional EBMs. Mathematically under an independence assumption, we can express the joint probability of the trajectory $\tau$ based on the set of concepts $\{c_1, \ldots, c_n\}$ in Eq. 2, and further reformulate the conditional terms by parameterizing $p(\boldsymbol{c}_i|\tau) \propto \left(\frac{p(\tau|\boldsymbol{c}_i)}{p(\tau)}\right)^\alpha$, as follows:

$$p(\tau|\boldsymbol{c}_1, \ldots, \boldsymbol{c}_n) \propto p(\tau, \boldsymbol{c}_1, \ldots, \boldsymbol{c}_n) = p(\tau) \prod_{i=1}^n p(\boldsymbol{c}_i|\tau), \tag{2}$$

$$\propto p(\tau) \prod_{i=1}^n \left(\frac{p(\tau|\boldsymbol{c}_i)}{p(\tau)}\right)^\alpha, \quad \text{with } p(\boldsymbol{c}_i|\tau) \propto \left(\frac{p(\tau|\boldsymbol{c}_i)}{p(\tau)}\right)^\alpha, \tag{3}$$

where $p(\boldsymbol{c}_i|\tau)$ can be interpreted as an implicit classifier (Ho & Salimans, 2022) and $\alpha$ serves as a weighting factor that modulates the influence of each concept on the overall trajectory distribution.

Then, the score function of the composed distribution can be derived directly from Eq. 3:

$$\nabla_\tau \log p(\tau|\boldsymbol{c}_1, \ldots, \boldsymbol{c}_n) = \nabla_\tau \log p(\tau) + \sum_{i=1}^n \alpha\big(\nabla_\tau \log p(\tau|\boldsymbol{c}_i) - \nabla_\tau \log p(\tau)\big). \tag{4}$$

Using the relationship between the score function of the distribution and noise (Bao et al., 2022), *i.e.*, $\epsilon_\theta(\tau_t, t) = -\sigma_\tau \nabla_\tau \log p(\tau)$, we can express the update rule for CDM with the $\epsilon$ parameterization :

$$\hat{\epsilon}(\tau_t, t, \boldsymbol{c}) = \epsilon_\theta(\tau_t, t) + \sum_{i=1}^n w_i\big(\epsilon_\theta(\tau_t, t, \boldsymbol{c}_i) - \epsilon_\theta(\tau_t, t)\big), \tag{5}$$

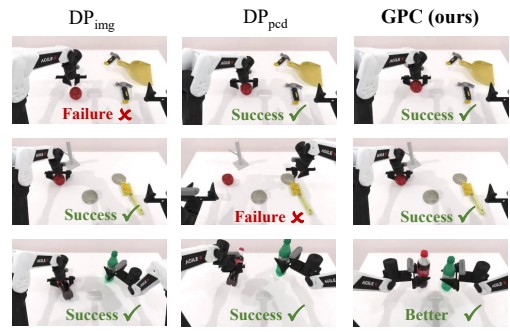

Figure 3: **Visualization results of different diffusion policies and the composed policy with GPC.** Our proposed GPC can be successful even when one part of the DP fails, and shows better performance when both parts of the DP work.

**Alg. 1: General Policy Composition Sampling**

**Input:** Pre-trained policies $\pi_1$, $\pi_2$, weights $w_1$, $w_2$ (*i.e.*, $1 - w_1$), policies' conditions $c_1$, $c_2$
1: **for** $w_1 = 0.0, 0.1, \ldots, 0.9, 1.0$ : // test-time searching
2: Initialize noise trajectory $\tau_N \sim \mathcal{N}(0, I)$ per action
3:     **for** $t = N, \ldots, 1$ : // denoising steps
4:         $s_1 \leftarrow \pi_1(\tau_t, t, c_1)$
5:         $s_2 \leftarrow \pi_2(\tau_t, t, c_2)$ # score estimation
6:         $\hat{s}_{\text{comp}} \leftarrow w_1 * s_1 + w_2 * s_2$ # score composition
7:         $\tau_{t-1} \leftarrow \alpha_t \tau_t + \beta_t \hat{s}_{\text{comp}} + \gamma_t$
8:     **Return:** Action trajectory $\tau_0$
9:     Evaluate SR and store in reward pools $R(w_1)$
**Return:** Optimal weights $w_1^* \leftarrow \arg\max_{w_1} R(w_1)$

Algorithm 1: **GPC Sampling.** Policies are combined via test-time score composition into a stronger policy.

where $\epsilon_\theta(\tau_t, t, c_i)/\epsilon_\theta(\tau_t, t)$ represents the noise estimation at time step $t$ for trajectory $\tau_t$ conditioned on the individual concept $c_i$ or without condition. The weights $w_i$ modulate the influence of each concept on the overall noise estimate. This formulation represents a generalization of the classifier-free guidance (CFG) (Ho & Salimans, 2022) technique commonly used in generative models.

## 5.2 GENERAL POLICY COMPOSITION

Based on the previous foundation, we can now apply the CDM to diffusion policy for robotic tasks. The joint probability distribution of the trajectory conditioned on different modes $c_i$ (*e.g.*, different visual modality input or different model architecture) can be expressed as follows:

$$p(\tau|c_1, \ldots, c_n) \propto p(\tau|c_1)p(\tau|c_2) \cdots p(\tau|c_n). \tag{6}$$

While one could in principle apply CFG sampling as in Eq. 5, our theoretical analysis in Sec. 4 shows that convex combinations of scores can provide a better functional objective and propagate stability through sampling dynamics. Motivated by this result, we construct our compositional policy by directly combining the score functions from multiple conditional diffusion policies via convex combination. This formulation not only inherits the stability guarantees established in theory but also enables flexible integration of diverse conditional information.

Formally, let $\hat{s}_{\text{comp}}(\tau_t, t, c)$ denotes the composed score, the update rule of GPC is defined as:

$$\hat{s}_{\text{comp}}(\tau_t, t, c) = \sum_{i=1}^{n} w_i s_\theta(\tau_t, t, c_i), \quad \text{with} \sum_{i=1}^{n} w_i = 1, \tag{7}$$

where $s_\theta(\tau_t, t, c_i)$ denotes the score estimate conditioned on concepts $c_i$ (*e.g.*, visual modality or policy architecture), and $w_i$ represents the weight of convex combination assigned to each concept, ensuring a balanced contribution from all source distributions in the final trajectory estimate.

This convex combination ensures that the composite score remains within the feasible convex hull of individual policies, preventing divergence toward extreme or unstable behaviors. Intuitively, the GPC formulation balances information from different conditions, yielding a more stable and coherent generative trajectory (*e.g.*, Fig. 3). GPC sampling process is shown in Alg. 1.

## 5.3 GPC WITH SUPERPOSITION

Apart from using the score convex combination, our GPC framework naturally connects to the principle of *superposition* (Skreta et al., 2024), which encompasses: **(i)** Logical OR corresponds to sampling from a mixture of distributions, which is implemented by weighting with the softmax function at each sampling time: $w_i^{1-t} = \text{softmax}\big(T \log p_t(\tau|c_i) + \ell\big)$, where $w_i^{1-t}$ determines the relative contribution of each policy's score in sampling time $t$, $T$ and $l$ are constants; **(ii)** Logical AND enforces agreement among policies, corresponding to the intersection of their distributions. This is achieved by solving a linear system to compute the weights such that $d \log p_t(\tau|c_i) = d \log p_t(\tau|c_j)$, ensuring consistency across different policies during sampling. In this work, we leverage these formulations to instantiate GPC with logical OR and AND operators as the application in Sec. 6.5.

Table 1: **Experiment results on Robomimic and PushT.** The table shows the success rate ↑. Our GPC yields a noticeable average improvement compared with the base policies.

| Method | Generative Mode | Model Type | Robomimic | | | PushT | |
|---|---|---|---|---|---|---|---|
| | | | Can | Lift | Square | PushT | Average |
| *Base Policies* | | | | | | | |
| Diffusion Policy (DP) | Diffusion | VA | 34.50 | 98.50 | 2.00 | 21.75 | 39.19 |
| Mamba Policy (MP) | Diffusion | VA | 5.00 | 98.50 | 3.00 | 12.06 | 29.64 |
| Flow Policy (FP) | Flow Matching | VA | 95.00 | 13.00 | 77.50 | 54.25 | 59.94 |
| Florence Policy-D | Diffusion | VLA | 61.50 | 97.00 | 46.50 | 40.00 | 61.25 |
| Florence Policy-F | Flow Matching | VLA | 89.00 | 98.50 | 88.50 | 39.38 | 78.84 |
| $\pi 0$ | Flow Matching | VLA | 96.50 | 99.00 | 92.50 | 57.69 | 86.42 |
| *Composed Policies via Convex Score Combination* | | | | | | | |
| DP+MP | Diffusion | VA & VA | 34.50 | 99.50 | 8.00 | 23.63 | 41.41 +2.22% |
| Florence-Policy-D+DP | Diffusion | VLA & VA | 62.50 | 100.00 | 61.50 | 43.06 | 66.76 +5.51% |
| Florence-Policy-D+MP | Diffusion | VLA & VA | 63.00 | 100.00 | 54.50 | 40.88 | 64.60 +3.35% |
| Florence-Policy-F+FP | Flow Matching | VLA & VA | 98.50 | 98.50 | 92.50 | 56.06 | 86.39 +7.55% |
| $\pi 0$+FP | Flow Matching | VLA & VA | 99.50 | 100.00 | 94.00 | 62.25 | 88.94 +2.52% |

Table 2: **Experiment results on RoboTwin with 6 diverse bimanual manipulation tasks.** GPC achieves an obvious increase with up to 7% improvement on the success rate.

| Method | Model Type | RoboTwin 2.0 | | | | | | |
|---|---|---|---|---|---|---|---|---|
| | | Hanging Mug | Open Laptop | Place Burger Fries | Put Object Cabinet | Stack Bowls Three | Turn Switch | Average |
| *Base Policies* | | | | | | | | |
| $DP_{img}$ | VA | 0.10 | 0.74 | 0.49 | 0.56 | 0.52 | 0.38 | 0.46 |
| $DP_{pcd}$ | VA | 0.21 | 0.93 | 0.72 | 0.71 | 0.64 | 0.71 | 0.65 |
| RDT | VLA | 0.13 | 0.69 | 0.46 | 0.32 | 0.47 | 0.30 | 0.40 |
| *Composed Policies via Convex Score Combination* | | | | | | | | |
| $DP_{img}$ + $DP_{pcd}$ | VA & VA | 0.23 | 0.93 | 0.78 | 0.82 | 0.71 | 0.71 | 0.70 +5% |
| RDT + $DP_{img}$ | VLA & VA | 0.18 | 0.80 | 0.57 | 0.59 | 0.66 | 0.38 | 0.53 +7% |
| RDT + $DP_{pcd}$ | VLA & VA | 0.36 | 0.94 | 0.83 | 0.78 | 0.73 | 0.71 | 0.72 +7% |

# 6 EXPERIMENT

We conduct experiments to investigate three key questions: **(i)** How does GPC perform in simulation and real-world experiments? **(ii)** How do different weight configurations influence the performance of GPC across various scenarios? **(iii)** How can the advantages of the composed DP be explained?

## 6.1 EXPERIMENT SETTINGS

**Environment Settings.** We evaluate on Robomimic (Mandlekar et al., 2022), which includes three manipulation tasks (Can, Lift, Square), PushT (Florence et al., 2021), and RoboTwin (Mu et al., 2025), a suite of dual-arm collaborative tasks where we select representative ones from versions 1.0 and 2.0 (Chen et al., 2025). We further perform four real-world experiments: Place Bottles, Hang Mug, Close Table, and Punch Holes, with the setups in Fig. 6. More details are in App. I.

**Baselines.** For Robomimic and PushT, we compare against three VA models: DP (Chi et al., 2023), Mamba Policy (MP) (Cao et al., 2025b), Flow Policy (FP, the flow matching version of DP), and three VLA models: Florence-based MDT (Reuss et al., 2024), Florence-based Flow-based MDT, and a revised $\pi 0$ (Black et al., 2024) built upon Florence VLM (Xiao et al., 2024). For RoboTwin, we adopt two VA models: DP, DP3 (Ze et al., 2024b), and a VLA model RDT (Liu et al., 2024a).

**Training and Testing Details.** Since GPC is training-free, we directly use pre-trained policies trained based on their original settings (more details are in App. I). Each setting is evaluated over 200 rollouts (100 for RoboTwin), and we report the average success rate (SR). For composition, we employ our GCP and search over weighting coefficients from 0.0 to 1.0 in steps of 0.1.

**Optimized weight searching (Optional).** Although GPC in principle allows arbitrary convex weights, our framework naturally suggests that the policy with the *stronger score* should receive a larger weight. This intuition is empirically validated by our findings in Sec. 6.3. Therefore, in practical use, once the relative strengths of the base policies are known, one can directly bias the stronger policy to have weight $>0.5$, instead of exhaustively searching over the entire interval $[0.1, 0.9]$. This simple heuristic both respects the underlying score-composition principle and significantly reduces the searching time.

Table 3: **Experiment results of our method under different composition configurations.** These results highlight GPC's versatility and the importance of weight tuning across policies.

| Scenario | Task | $DP_{img}$ | $DP_{pcd}$ | Weight Scheduling in GPC | | | | | | | | | $\Delta$ |
|---|---|---|---|---|---|---|---|---|---|---|---|---|---|
| | | | | 0.1* | 0.2 | 0.3 | 0.4 | 0.5 | 0.6 | 0.7 | 0.8 | 0.9 | |
| *Both Policies Perform Well* | Empty Cup Place | 0.42 | 0.62 | 0.70 | **0.86** | 0.84 | **0.86** | 0.84 | 0.84 | 0.76 | 0.68 | 0.61 | +24% |
| | Dual Bottles Pick (Hard) | 0.49 | 0.64 | 0.69 | 0.63 | **0.71** | 0.66 | 0.64 | 0.65 | 0.63 | 0.56 | 0.58 | +7% |
| | Shoe Place | 0.37 | 0.36 | 0.47 | 0.52 | 0.56 | 0.59 | **0.60** | 0.59 | 0.59 | 0.53 | 0.41 | +23% |
| *Both Policies Perform Bad* | Dual Shoes Place | 0.08 | **0.23** | 0.19 | 0.17 | 0.19 | 0.20 | 0.20 | 0.17 | 0.16 | 0.14 | 0.09 | +0% |
| | Pick Apple Messy | 0.05 | **0.26** | 0.25 | 0.17 | 0.21 | 0.15 | 0.13 | 0.08 | 0.08 | 0.06 | 0.08 | +0% |
| *Policy A > Policy B* | Dual Bottles Pick (Easy) | 0.77 | 0.36 | 0.52 | 0.64 | 0.70 | 0.75 | 0.82 | 0.81 | 0.80 | **0.85** | 0.80 | +8% |
| *Policy A < Policy B* | Block Hammer Beat | 0.00 | **0.76** | 0.61 | 0.3 | 0.18 | 0.15 | 0.12 | 0.07 | 0.00 | 0.00 | 0.00 | +0% |

*: The number set $\{0.1, ..., 0.9\}$ denotes the weight of $DP_{img}$ (*i.e.*, $w_1$), corresponding to the noise estimation of GPC as $\hat{\epsilon}_{\mathcal{M}*} = w_1 * \epsilon_{DP_{img}} + w_2 * \epsilon_{DP_{pcd}}$. When $w_1$ equals to 0.0 and 1.0, GPC degenerates into $DP_{pcd}$ and $DP_{img}$, respectively.

(a) Sample Distribution under Different Modality   (b) Sample Distribution under Different Models

Figure 4: **Visual analysis of GPC under different compositions.** GPC generalizes across (a) modalities and (b) architectures, with appropriate weighting yielding accurate distributions with better SR than individual policies.

Figure 5: **Sample distribution through execution time.** GPC yields more coherent distributions than baselines.

## 6.2 Main Results: GPC Across Architectures and Modalities

**Simulation Results.** Our results demonstrate that GPC is broadly applicable across both diffusion- and flow-based policies, and works under a range of *general* settings: *(i) Same input modality, different architectures.* GPC successfully composes policies trained on the same modality but with different network architectures. For instance, in Tab. 1, combining two VA policies (DP+MP) yields a noticeable average improvement of +2.22% over the base policies, while combining a VA and VLA model (Florence-D+DP) achieves a larger increase of +5.51%. *(ii) Different modalities, similar architectures.* GPC also supports the integration of heterogeneous modalities. In Tab. 2, combining RGB-based and point cloud-based DPs ($DP_{img}+DP_{pcd}$) improves the average SR from 0.46/0.65 to 0.70 (+5%), confirming that convex score composition can exploit complementary information even within the same sensory domain. *(iii) Different modalities, different architectures.* GPC enables flexible integration across modalities and architectures. For example, combining a VLA model with a VA policy (RDT+$DP_{pcd}$) produces consistent improvements, raising the average SR to +7% compared to $DP_{pcd}$, and surprisingly, +32% compared to RDT itself.

**Real-world Results.** In real-world evaluations (in Tab. 5), GPC shows consistently stronger performance than single-policy baselines. For instance, in the Clean Table task it achieves 14/20 successes, surpassing base policies. Similarly, it delivered gains in Place Bottles (13/20 *vs.* 7/20 and 11/20).

Overall, to answer question one, across diverse tasks and benchmarks, GPC consistently improves performance, with an average increase of up to +7.55% on Robomimic & PushT, +7% on RoboTwin, and +10% in real-world tasks. These results validate that convex score composition provides a robust and general principle for composing policies, regardless of model type or input modality.

## 6.3 Influence of Weight Configurations on GPC Performance

To analyze the second question, we evaluate GPC performance across multiple tasks under different weight configurations in Tab. 3. Several findings are summarized:

🔍 **Finding 1: When both policies have moderate accuracy (*e.g.*, >30%), GPC often achieves higher accuracy under appropriate weight configurations compared to base policies.** For in-

Table 4: **Results of GPC with superposition,** highlighting performance increase by strong compositional operators.

| Method | Robomimic | | | PushT | |
|---|---|---|---|---|---|
| | Can | Lift | Square | PushT | Average |
| *Base Policies* | | | | | |
| Diffusion Policy (DP) | 34.50 | 98.50 | 2.00 | 21.75 | 39.19 |
| Mamba Policy (MP) | 5.00 | 98.50 | 3.00 | 12.06 | 29.64 |
| Florence Policy-D | 61.50 | 97.00 | 46.50 | 40.00 | 61.25 |
| *Composed Policies via Logical AND Composition* | | | | | |
| DP+MP | 84.00 | 99.50 | 48.00 | 28.18 | 64.92 +25.73% |
| Florence-Policy-D+DP | **90.50** | **100.00** | **90.00** | 36.31 | 79.20 +17.95% |
| Florence-Policy-D+MP | 83.00 | **100.00** | **90.00** | 37.38 | 77.60 +16.35% |
| *Composed Policies via Logical OR Composition* | | | | | |
| DP+MP | 82.50 | 99.50 | 44.00 | 29.13 | 63.78 +24.59% |
| Florence-Policy-D+DP | 83.50 | **100.00** | 89.00 | 37.87 | 77.59 +16.34% |
| Florence-Policy-D+MP | 86.50 | **100.00** | 86.50 | **38.44** | 77.86 +16.61% |

Table 5: **Real-world experiment results,** demonstrating the effectiveness of GPC.

| Method | Place Bottles | Hang Mug | Clean Table | Punch Holes |
|---|---|---|---|---|
| $DP_{img}$ | 7/20 | 5/20 | 12/20 | 7/20 |
| $DP_{pcd}$ | 11/20 | 6/20 | 7/20 | 6/20 |
| **GPC (ours)** | **13/20** | **7/20** | **14/20** | **9/20** |

Figure 6: **Real-world setup and results.**

Table 6: Comparison of training/finetuning time vs. GPC weight search. $T_{eval} = N_{rollout} \times T_{per\_rollout}$. For RoboMimic, $T_{eval}^{RoboMimic} \approx 200 * 5s = 0.27$ hr. For real-world, $T_{eval}^{Real} \approx 20 * 30s = 0.17$ hr.

| Method | Setting | Time cost |
|---|---|---|
| Training from scratch | 1M+ demos, N GPUs | 14d (OpenVLA 7B), 30d (RDT 1B) |
| Finetuning | 100 demos, 1 GPU | >5h (DP 200M+), >8h (RDT 1B) |
| GPC (full search) | 9 weights ($w = 0.1:0.9$) | $9\,T_{eval}$ ($\sim$ 2.5hrs) |
| GPC (optimized) | 4 weights ($w_{strong} \in [0.6, 0.9]$) | $4\,T_{eval}$ ($\sim$ 1hr) |

Table 7: Per–action-chunk inference latency in RoboMimic. The overhead of GPC is modest and purely computational.

| Method | Time per chunk (s) |
|---|---|
| DP | 0.09 |
| Florence-Policy-D | 0.06 |
| GPC (DP + FP-D) | 0.13 |

stance, in the Empty Cup Place task, $DP_{img}$ and $DP_{pcd}$ achieve 0.42 and 0.62, respectively, while GPC peaks at 0.86 (+24%) with $w_1$=0.4, surpassing both unimodal DPs. This improvement reflects the composition of diffusion scores capturing a more generalized distribution that reduces the reliance on specific conditions, consistent with the theoretical advantages of compositional models.

🔍 **Finding 2: When one policy has significantly lower accuracy, GPC struggles to surpass the highest accuracy of the better-performing base policies.** For example, in the Pick Apple Messy task, $DP_{pcd}$ achieves 0.26 and $DP_{img}$ achieves only 0.05. GPC peaks at 0.25, falling short of $DP_{pcd}$. This suggests that low-accuracy scores from weaker modalities can significantly impact the joint distribution, diminishing the overall performance of the composed policy.

🔍 **Finding 3: The improvement of GPC is always maximized when the better-performing base policy holds a larger weight in GPC.** For instance, in Dual Bottles Pick (Easy), where $DP_{img}$ achieves 0.77, GPC reaches 0.85 with $w_1$=0.8, leveraging the stronger DP effectively. This highlights the necessity of assigning higher weights to the better-performing distribution to maximize the effectiveness of GPC, leading the composed policy toward consensus.

These findings highlight GPC's versatility in leveraging the strengths of different conditions and the importance of appropriately tuning weights to each policy's performance.

### 6.4 TIME AND COMPUTATIONAL EFFICIENCY OF GPC

GPC introduces two main sources of extra cost compared to using a single base policy: (i) additional weight search evaluations (Snell et al., 2024), and (ii) increased inference cost per action chunk.

**Weight search as repeated evaluations.** During weight search, we fix a candidate weight configuration, run $N$ rollouts to estimate the SR, and then move to the next configuration. Thus, the total search cost is $T_{search} = N_{search} * T_{eval}$, $T_{eval} = N_{rollout} * T_{per\_rollout}$. As shown in Tab. 6, sweeping $w$ from 0.1 to 0.9 leads to $N_{search} = 9$ and about 2.5 hours. With our optimized searching method, we can restrict the search to $w_{strong} \in [0.6, 0.9]$, reducing to $N_{cfg} = 4$ ($\sim$ 1 hour). Compared to days of training from scratch or hours of finetuning, this makes GPC a highly efficient alternative.

**Inference-time overhead.** During rollout, GPC incurs extra compute from querying multiple base policies per action chunk. Tab. 7 reports the measured inference latency in RoboMimic: the GPC composition increases latency from 0.09 s to 0.13 s per chunk, which is modest in practice. This overhead is purely computational and can be further reduced with stronger hardware, optimized inference runtimes, or future engineering improvements (*e.g.*, model compression (Høeg et al., 2024) or distillation (Liu et al., 2026)).

Table 8: GPC can be applied with different action-chunk lengths and infer time steps. Results are in Robomimic.

| Method | Setting | Success Rate |
|---|---|---|
| DP | DDPM, chunk 8, 5 steps | 0.50 |
| Florence-Policy-D | DDIM, chunk 16, 10 steps | 0.53 |
| **GPC** (DP+FP-D) | DDIM, chunk 16, 10 steps | **0.66** |

Table 9: GPC with three base policies on RoboMimic.

| Method | Can | Lift | Square |
|---|---|---|---|
| Flow Policy | 0.95 | 0.13 | 0.77 |
| Florence-Policy-F | 0.89 | 0.98 | 0.88 |
| $\pi_0$ | 0.61 | 0.96 | 0.92 |
| **GPC** (best 2-policy) | 0.99 | **1.00** | **0.94** |
| **GPC** (FP+FP-F+$\pi_0$) | **1.00** | **1.00** | **0.94** |

## 6.5 COMPREHENSIVE ANALYSIS OF GPC EFFECTIVENESS

**Analysis on GPC's Superiority via Visualization.** For the third question, Fig. 4 illustrates how GPC improves sample distributions under different settings: *(i) GPC under different modalities.* In Fig. 4(a), $DP_{img}$ and $DP_{pcd}$ learn distinct distributions. By adjusting the convex weights, GPC adapts smoothly between them. This demonstrates how GPC leverages knowledge from different modalities to form a more complete distribution. *(ii) GPC under different architectures.* In Fig. 4(b), both Florence and FlowP learn broadly similar distributions, yet each exhibits localized biases. Through convex composition, GPC expands coverage and enhances precision. This shows that even when base models learn similar representations, GPC refines their alignment and achieves stronger results. Overall, these visualizations confirm that GPC generalizes across modalities and architectures, with appropriate weighting yielding broader and more accurate distributions than individual policies.

**Analysis on Execution-Time Sample Distributions.** Fig. 5 shows the evolution of execution-time sample distributions. Baselines $DP_{img}$ and $DP_{pcd}$ produce scattered or noisy patterns, particularly in later stages, indicating instability and higher variance. In contrast, GPC yields coherent and concentrated distributions, ensuring greater stability and mitigating error accumulation during execution.

**Experiment Results on GPC with Superposition.** We further evaluate GPC under superposition settings. As shown in Tab. 4, composing DP and MP with logical AND boosts the SR to 64.92 (+25.73%), while Florence-D + DP under logical OR reaches 77.59 (+16.34%). These results highlight the potential of superposition to amplify policy performance through stronger composition operators. However, superposition also has clear limitations. It is not directly applicable to flow-based models, and the requirement to recompute weights at every step increases inference cost.

**GPC with heterogeneous inference steps and chunk sizes.** GPC is also flexible with different inference steps and chunk sizes. For diffusion steps, we simply pick a unified sampler and number of inference steps at test time and run all policies under this common configuration, *e.g.*, the results in RoboTwin with DP+RDT. For action-chunk mismatch (assume $H_A \geq H_B$), we sample a shared noise trajectory of length $H_A$, apply policy B only on the first $H_B$ steps, and take a convex combination of scores on the overlap while keeping the tail from policy A. Results in Tab. 8 confirm that this heterogeneous-chunk composition also yields clear gains.

**GPC with multiple base policies.** The score-composition rule of GPC directly extends to three or more base policies via a convex combination of their scores. Tab. 9 reports results with three policies on RoboMimic. Three-policy GPC either matches or improves upon the best two-policy configuration, and consistently outperforms the individual base policies. This shows that GPC scales beyond pairwise composition and can effectively leverage diversity across multiple pretrained policies.

## 7 DISCUSSION

**Limitations.** Our GPC demonstrates clear effectiveness across a wide range of experiments. Despite this strength, certain limitations remain. First, test-time weight search is restricted by a fixed discretization, which may overlook optimal values; future work could explore adaptive or automatic search strategies. Second, we mainly study dual/triple-policy composition, while scaling to more policies increases computation. Addressing this may require feature sharing or compact representations to enable efficient multi-policy integration.

**Conclusion.** We introduced General Policy Composition, a training-free framework that improves robotic control by combining the distributional scores of pre-trained policies. Our theoretical analysis establishes that convex score composition leads to step-wise and trajectory-level improvements, while our experiments on diverse benchmarks and real-world setups confirm consistent performance gains. GPC is simple, versatile, and widely applicable, providing a foundation for future research in policy composition as a means to enhance performance without additional training resources.

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

## APPENDIX

## A  ASSUMPTIONS, NOTATION, AND PRELIMINARY FACTS

**Dynamics.**  Sampling is modeled as an ODE/SDE:

$$\dot{x}(t) \;=\; F\big(t,\, x(t),\, s(t, x(t))\big), \qquad t \in [0, T], \quad x(0) \sim \mu_0, \tag{8}$$

where $x(t) \in \mathbb{R}^d$, the score $s : [0, T] \times \mathbb{R}^d \to \mathbb{R}^d$, and $F : [0, T] \times \mathbb{R}^d \times \mathbb{R}^d \to \mathbb{R}^d$ represents the transformed score due to noise schedule, parameterization, or solver.

**Oracle vs. realized flow.**  We compare the oracle trajectory $x^*(t)$ solving equation 8 with $s^*$, and the realized trajectory $x_{\hat{s}}(t)$ solving equation 8 with $\hat{s}$; both share the same initial conditions.

**Score error.**

$$\Delta_s(t, x) := \hat{s}(t, x) - s^*(t, x).$$

**Score error bound assumption.**  We assume:

There exists a nonnegative function $\kappa(t)$ such that for all $x$,

$$\|\Delta_s(t, x)\| \leqslant \kappa(t).$$

**Regularity assumptions.**  We assume:

(A1) **Lipschitz of $F$.** (Sohrab, 2003; Thomson et al., 2008) For a.e. $t \in [0, T]$, there exist integrable $L_x, L_s \geqslant 0$ such that, for all $x, y, s, r$,

$$\|F(t, x, s) - F(t, y, r)\| \leqslant L_x(t)\|x - y\| + L_s(t)\|s - r\|.$$

(A2) **Score regularity.** $s^*(t, \cdot)$ and $\hat{s}(t, \cdot)$ are locally Lipschitz on the tube visited by the flows, with (time-dependent) moduli $\Lambda^*(t), \hat{\Lambda}(t) \in L^1([0, T])$ and at most linear growth.

**Absolute continuity and norm derivative.**  If $e : [0, T] \to \mathbb{R}^d$ is absolutely continuous, then $e'(t)$ exists a.e., $e(t) = e(0) + \int_0^t e'(\tau)\, d\tau$, and $\phi(t) := \|e(t)\|$ is absolutely continuous with

$$\phi'(t) \;\leqslant\; \|e'(t)\| \quad \text{for a.e. } t. \tag{9}$$

We give a complete proof in §E.

**Why Lipschitz is reasonable.**  For probability-flow ODEs of score-based models, $F$ is affine in $s$ (*e.g.*, reverse ODE (Song et al., 2020b): $F = f(t, x) - g(t)^2 s(t, x)$), so $L_s(t) = g(t)^2$ (known, bounded on finite $T$). The dependence on $x$ comes via $f(t, x)$ (often smooth) and through $s$; with networks on compact tubes, local Lipschitz holds and yields finite Lipschitz moduli $\Lambda^*(t), \hat{\Lambda}(t)$. These are standard in stability analyses of neural ODEs and probability-flow ODEs.

## B    PROOF OF PROPOSITION 4.1 (SINGLE-STEP CONVEX IMPROVEMENT)

**Statement restated.**    Conditioning on $(t, x_t)$, suppose we have two estimators

$$\varepsilon_i = s^*(t, x_t) + b_i(t, x_t) + \eta_i, \qquad i = \{1, 2\},$$

where $s^*(t, x_t)$ is the true score, $b_i$ are deterministic biases, and $\eta_i$ are zero-mean random noises. For $w \in [0, 1]$, define the convex combination

$$\varepsilon(w) = w\varepsilon_1 + (1 - w)\varepsilon_2, \qquad Q(w) := \mathbb{E}_\eta \|\varepsilon(w) - s^*(t, x_t)\|^2. \tag{10}$$

**Notation.**    This abstraction (equation 10) unifies different modeling paradigms:

- When noise is present, the estimator can be viewed as the output of a diffusion model, and the residual term $\eta$ plays the role of the diffusion component in the time-reversed stochastic dynamics (*e.g.*, a reverse-time ODE).

- When the noise term vanishes (*i.e.*, $\eta = 0$), the formulation reduces to a deterministic transport setting, which is the flow matching case.

The role of $\eta_i$ is analogous to the stochastic noise introduced in the diffusion forward process (*e.g.*, Gaussian perturbations of the clean sample), ensuring that each estimator $\varepsilon_i$ remains random even when $(t, x_t)$ is fixed. All expectations in Proposition 4.1 are therefore taken with respect to the joint distribution of $(\eta_1, \eta_2)$. and the randomness is solely due to $(\eta_1, \eta_2)$. We write expectations as

$$\mathbb{E}[\cdot] \equiv \mathbb{E}_\eta[\cdot] \equiv \mathbb{E}_{\eta_1, \eta_2}[\cdot \mid x_t, t].$$

**Goal.**    Show that $Q(w)$ is a convex quadratic, derive its coefficients, the minimizer $w^\star$, the minimum value, and conditions for improvement over the endpoints $w = 0, 1$.

**Decomposition.**    Subtracting $s^*(t, x_t)$, we write

$$\varepsilon(w) - s^* = u(w) + v(w),$$

where

$$u(w) = wb_1 + (1 - w)b_2, \qquad v(w) = w\eta_1 + (1 - w)\eta_2.$$

Hence

$$\begin{aligned} Q(w) &= \mathbb{E}\|u(w) + v(w)\|^2 \\ &= \|u(w)\|^2 + 2\,\mathbb{E}\langle u(w), v(w)\rangle + \mathbb{E}\|v(w)\|^2. \end{aligned} \tag{11}$$

Since $\mathbb{E}[\eta_i] = 0$, the cross term vanishes, leaving

$$Q(w) = \|u(w)\|^2 + \mathbb{E}\|v(w)\|^2.$$

**Bias contribution.**    Expanding $\|u(w)\|^2$ gives

$$\|u(w)\|^2 = \|b_2 + w(b_1 - b_2)\|^2 = \|b_2\|^2 + 2w\langle b_2, b_1 - b_2\rangle + w^2\|b_1 - b_2\|^2.$$

**Noise contribution.**    Expanding $\mathbb{E}\|v(w)\|^2$ gives

$$\begin{aligned} \mathbb{E}\|v(w)\|^2 &= \mathbb{E}\|w\eta_1 + (1 - w)\eta_2\|^2 \\ &= w^2\,\mathbb{E}\|\eta_1\|^2 + (1 - w)^2\,\mathbb{E}\|\eta_2\|^2 + 2w(1 - w)\,\mathbb{E}\langle\eta_1, \eta_2\rangle. \end{aligned}$$

**Quadratic form.**    Combining the two contributions, $Q(w)$ is a quadratic function

$$Q(w) = Aw^2 + Bw + C,$$

where

$$A = \|b_1 - b_2\|^2 + \mathbb{E}\|\eta_1\|^2 + \mathbb{E}\|\eta_2\|^2 - 2\mathbb{E}\langle\eta_1, \eta_2\rangle,$$
$$B = 2\langle b_2, b_1 - b_2\rangle - 2\mathbb{E}\|\eta_2\|^2 + 2\mathbb{E}\langle\eta_1, \eta_2\rangle,$$
$$C = \|b_2\|^2 + \mathbb{E}\|\eta_2\|^2.$$

**Convexity.** By Cauchy–Schwarz,

$$\mathbb{E}\langle \eta_1, \eta_2 \rangle \leqslant \sqrt{\mathbb{E}\|\eta_1\|^2 \, \mathbb{E}\|\eta_2\|^2},$$

so

$$A \geqslant (\sqrt{\mathbb{E}\|\eta_1\|^2} - \sqrt{\mathbb{E}\|\eta_2\|^2})^2 + \|b_1 - b_2\|^2 \geqslant 0.$$

Thus $Q(w)$ is convex, and strictly convex unless both biases and noises coincide.

**Minimizer.** If $A > 0$, the unique minimizer is

$$w^\star = -\frac{B}{2A} = \frac{\mathbb{E}\|\eta_2\|^2 - \mathbb{E}\langle \eta_1, \eta_2 \rangle - \langle b_2, b_1 - b_2 \rangle}{\|b_1 - b_2\|^2 + \mathbb{E}\|\eta_1\|^2 + \mathbb{E}\|\eta_2\|^2 - 2\mathbb{E}\langle \eta_1, \eta_2 \rangle}.$$

The minimum value is

$$Q(w^\star) = C - \frac{B^2}{4A}.$$

**Endpoint comparison.** At $w = 0, 1$, we have

$$Q(0) = C, \qquad Q(1) = A + B + C.$$

The gaps are

$$Q(0) - Q(w^\star) = \frac{B^2}{4A} \geqslant 0, \qquad Q(1) - Q(w^\star) = \frac{(2A+B)^2}{4A} \geqslant 0,$$

with strict inequality unless $B = 0$ or $2A + B = 0$.

**Special Case I: Unbiased case.** If $b_1 = b_2 = 0$, then

$$A = \mathbb{E}\|\eta_1\|^2 + \mathbb{E}\|\eta_2\|^2 - 2\mathbb{E}\langle \eta_1, \eta_2 \rangle, \quad B = -2\mathbb{E}\|\eta_2\|^2 + 2\mathbb{E}\langle \eta_1, \eta_2 \rangle, \quad C = \mathbb{E}\|\eta_2\|^2,$$

with $A \geqslant 0$ and $A > 0$ unless $\eta_1, \eta_2$ are perfectly correlated with identical second moments. For $A > 0$, the unique minimizer is

$$w^\star = \frac{\mathbb{E}\|\eta_2\|^2 - \mathbb{E}\langle \eta_1, \eta_2 \rangle}{\mathbb{E}\|\eta_1\|^2 + \mathbb{E}\|\eta_2\|^2 - 2\mathbb{E}\langle \eta_1, \eta_2 \rangle},$$

and the endpoint gaps are

$$Q(0) - Q(w^\star) = \frac{B^2}{4A} = \frac{\left(\mathbb{E}\|\eta_2\|^2 - \mathbb{E}\langle \eta_1, \eta_2 \rangle\right)^2}{\mathbb{E}\|\eta_1\|^2 + \mathbb{E}\|\eta_2\|^2 - 2\mathbb{E}\langle \eta_1, \eta_2 \rangle} > 0$$

whenever $\mathbb{E}\langle \eta_1, \eta_2 \rangle \neq \mathbb{E}\|\eta_2\|^2$, and

$$Q(1) - Q(w^\star) = \frac{(2A + B)^2}{4A} = \frac{\left(\mathbb{E}\|\eta_1\|^2 - \mathbb{E}\langle \eta_1, \eta_2 \rangle\right)^2}{\mathbb{E}\|\eta_1\|^2 + \mathbb{E}\|\eta_2\|^2 - 2\mathbb{E}\langle \eta_1, \eta_2 \rangle} > 0$$

whenever $\mathbb{E}\langle \eta_1, \eta_2 \rangle \neq \mathbb{E}\|\eta_1\|^2$. Thus $Q(w^\star) < \min\{Q(0), Q(1)\}$ whenever $\eta_1, \eta_2$ are not perfectly correlated.

**Special Case II: No-noise (deterministic) case with bias** Assume $\eta_1 = \eta_2 = 0$ (deterministic estimators with bias). Then

$$Q(w) = \|wb_1 + (1 - w)b_2\|^2 = \|b_2 + w(b_1 - b_2)\|^2, \qquad w \in [0, 1].$$

Write $Q(w) = \alpha w^2 + 2\beta w + \gamma$ with

$$\alpha = \|b_1 - b_2\|^2, \qquad \beta = \langle b_2, b_1 - b_2 \rangle, \qquad \gamma = \|b_2\|^2.$$

If $b_1 \neq b_2$ then $\alpha > 0$ and the unconstrained minimizer is $w_{\mathbb{R}}^\star = -\beta/\alpha$, giving

$$\min_{w \in \mathbb{R}} Q(w) = \gamma - \frac{\beta^2}{\alpha}.$$

Hence the endpoint gaps are

$$Q(0) - \min Q = \frac{\beta^2}{\alpha} = \frac{\langle b_2, b_1 - b_2 \rangle^2}{\|b_1 - b_2\|^2} \geqslant 0, \qquad Q(1) - \min Q = \alpha + 2\beta + \frac{\beta^2}{\alpha} = \frac{(\alpha + \beta)^2}{\alpha} \geqslant 0.$$

Therefore, if $w_{\mathbb{R}}^\star \in (0, 1)$ (so the constrained minimizer over $[0, 1]$ equals the unconstrained one), we have

$$Q(w^\star) = \min Q < \min\{Q(0), Q(1)\},$$

with strict inequalities unless $\beta = 0$ (for $Q(0)$) or $\alpha + \beta = 0$ (for $Q(1)$). If $w_{\mathbb{R}}^\star \notin (0, 1)$, the constrained minimizer lies at an endpoint and no strict improvement over both endpoints is possible. If $b_1 = b_2$, then $\alpha = 0$ and $Q(w) \equiv \|b_1\|^2$ for all $w$.

**Remark on strict improvement.** Strict improvement requires the unconstrained optimum to lie within $(0, 1)$. If one estimator significantly outperforms the other, the optimal weight lies on the boundary, recovering the best single-estimator performance.

## C Proof of Proposition 4.2 (Score-to-Sample stability)

**Overview before proof.** The goal of the score-to-sample stability result is to show how errors in the score estimation translates into deviations of the generated trajectory. Formally, we view sampling as an ODE of the form $\dot{x}(t) = F(t, x(t), s(t, x(t)))$, where the score $s$ directly drives the dynamics, and $F$ represents the transformed output after scheduler, parameterization, or solver. Replacing the oracle score $s^*$ with an estimator $\hat{s}$ perturbs this vector field, and the resulting trajectory deviation can be quantified.

The proof proceeds by analyzing the trajectory difference $e(t) = x_{\hat{s}}(t) - x^*(t)$. Its derivative naturally splits into two terms: a Lipschitz growth component proportional to $\|e(t)\|$, and a forcing component proportional to the score error $\|\hat{s} - s^*\|$. This reduces the problem to a standard stability inequality for ODEs. Applying Grönwall's inequality (Gronwall, 1919; Bellman, 1943) then yields a trajectory-level bound expressed in terms of the integrated score error.

Finally, this stability guarantee connects back to Proposition 4.1: since convex composition strictly improves score estimation at the single-step level, the bound implies that the composed policy inherits a strictly tighter trajectory deviation bound. This prepares the ground for Corollary 4.1, which consolidates the results into a trajectory-level performance guarantee for GPC.

**Statement restated.** Let $x^*(t)$ and $x_{\hat{s}}(t)$ solve

$$\dot{x}^*(t) = F(t, x^*(t), s^*(t, x^*(t))), \qquad \dot{x}_{\hat{s}}(t) = F(t, x_{\hat{s}}(t), \hat{s}(t, x_{\hat{s}}(t))),$$

with the same $x(0)$. Under (A1)–(A2), with

$$\tilde{L}(t) := L_x(t) + L_s(t)\hat{\Lambda}(t),$$

we have for all $T \in [0, T]$:

$$\|x_{\hat{s}}(T) - x^*(T)\| \leq \int_0^T \exp\left(\int_t^T \tilde{L}(\tau)d\tau\right) L_s(t) \|\Delta_s(t, x^*(t))\| \, dt. \tag{12}$$

Taking expectation, applying Cauchy–Schwarz and Jensen and using the assumption of score error bound, we obtain

$$\mathbb{E}\|x_{\hat{s}}(T) - x^*(T)\| \leq \left(\int_0^T e^{2\int_t^T \tilde{L}(\tau)\,d\tau} L_s(t)^2 \, dt\right)^{1/2} \left(\int_0^T \kappa(t)^2 \, dt\right)^{1/2}, \tag{13}$$

where $\kappa(t)$ is a nonnegative function $\kappa(t)$ such that for all $x$,

$$\|\Delta_s(t, x)\| \leq \kappa(t).$$

(*i.e.*, Score error bound assumption). Please see the detailed proof as follows.

### A. Absolute continuity and the error differential inequality

Let $e(t) := x_{\hat{s}}(t) - x^*(t)$. we have

$$e'(t) = F(t, x_{\hat{s}}(t), \hat{s}(t, x_{\hat{s}}(t))) - F(t, x^*(t), s^*(t, x^*(t))) \quad \text{for a.e. } t. \tag{14}$$

Insert and subtract two intermediate terms to separate $x$ and $s$ contributions:

$$\|e'(t)\| \leq \left\|F(t, x_{\hat{s}}, \hat{s}(t, x_{\hat{s}})) - F(t, x_{\hat{s}}, \hat{s}(t, x^*))\right\| \tag{15}$$

$$+ \left\|F(t, x_{\hat{s}}, \hat{s}(t, x^*)) - F(t, x_{\hat{s}}, s^*(t, x^*))\right\| \tag{16}$$

$$+ \left\|F(t, x_{\hat{s}}, s^*(t, x^*)) - F(t, x^*, s^*(t, x^*))\right\|. \tag{17}$$

By (A1), the first (15) and second (16) equations are bounded by $L_s(t)\|\hat{s}(t, x_{\hat{s}}) - \hat{s}(t, x^*)\|$ and $L_s(t)\|\hat{s}(t, x^*) - s^*(t, x^*)\|$, respectively; the third equation 17 is bounded by $L_x(t)\|x_{\hat{s}} - x^*\| = L_x(t)\|e(t)\|$. Using the $x$-Lipschitzness of $\hat{s}$ from (A2),

$$\|\hat{s}(t, x_{\hat{s}}) - \hat{s}(t, x^*)\| \leqslant \hat{\Lambda}(t)\, \|e(t)\|.$$

Therefore,

$$\|e'(t)\| \;\leqslant\; \underbrace{\Big(L_x(t) + L_s(t)\hat{\Lambda}(t)\Big)}_{:=\tilde{L}(t)}\, \|e(t)\| \;+\; L_s(t)\, \|\Delta_s(t, x^*(t))\|. \tag{18}$$

## B. From $\|e'(t)\|$ to $\phi'(t)$

Let $\phi(t) := \|e(t)\|$. By equation 9, $\phi$ is absolutely continuous and

$$\phi'(t) \leqslant \|e'(t)\| \quad \text{for a.e. } t.$$

Combining with equation 18 gives the scalar differential inequality

$$\phi'(t) \;\leqslant\; \tilde{L}(t)\, \phi(t) \;+\; L_s(t)\, \|\Delta_s(t, x^*(t))\| \quad \text{for a.e. } t, \qquad \phi(0) = 0. \tag{19}$$

## C. Grönwall (integrating factor) and the pathwise bound

Define $A(t) := \int_0^t \tilde{L}(\tau)\, d\tau$ and $g(t) := e^{-A(t)}\phi(t)$. Then a.e.

$$g'(t) = e^{-A(t)}\big(\phi'(t) - \tilde{L}(t)\phi(t)\big) \;\leqslant\; e^{-A(t)}L_s(t)\, \|\Delta_s(t, x^*(t))\|.$$

Integrate from 0 to $T$; since $\phi(0) = 0$ (same initial conditions) we have $g(0) = 0$:

$$g(T) \;\leqslant\; \int_0^T e^{-A(t)}L_s(t)\, \|\Delta_s(t, x^*(t))\|\, dt.$$

Multiply by $e^{A(T)}$:

$$\phi(T) \;=\; e^{A(T)}g(T) \;\leqslant\; \int_0^T e^{A(T)-A(t)}L_s(t)\, \|\Delta_s(t, x^*(t))\|\, dt.$$

Since $A(T) - A(t) = \int_t^T \tilde{L}(\tau)\, d\tau$, the bound

$$\phi(T) = \|e(T)\| \;\leqslant\; \int_0^T \exp\Big(\int_t^T \tilde{L}(\tau)d\tau\Big) L_s(t)\, \|\Delta_s(t, x^*(t))\|\, dt \tag{20}$$

follows, which is exactly equation 12.

## D. Expectation and a Readable Upper Bound

We first take expectations of the pathwise bound equation 20:

$$\mathbb{E}\|e(T)\| \;=\; \mathbb{E}\left[\int_0^T e^{\int_t^T \tilde{L}(\tau)\, d\tau}\, L_s(t)\, \|\Delta_s(t, x^*(t))\|\, dt\right].$$

**Notation.** The expectation $\mathbb{E}[\cdot]$ is taken over the randomness of the initial conditions. This **already provides a valid (and tight) expected bound.** In the following we present a **slightly looser but cleaner** form by applying classical inequalities, which is easier to read and to apply in practice.

By Tonelli's theorem (non-negative integrand) (Fubini, 1907; Tonelli, 1909):

$$= \int_0^T e^{\int_t^T \tilde{L}(\tau)\, d\tau}\, L_s(t)\, \mathbb{E}\|\Delta_s(t, x^*(t))\|\, dt.$$

Apply Cauchy–Schwarz (Cauchy, 1821) in $L^2([0, T])$:

$$\int_0^T W(t)\, \mathbb{E}\|\Delta_s(t, x^*(t))\|\, dt \;\leqslant\; \Big(\int_0^T W(t)^2\, dt\Big)^{1/2}\Big(\int_0^T \big(\mathbb{E}\|\Delta_s(t, x^*(t))\|\big)^2\, dt\Big)^{1/2},$$

where
$$W(t) := e^{\int_t^T \tilde{L}(\tau)\, d\tau}\, L_s(t).$$

Use Jensen (Jensen, 1906) on $(\mathbb{E}\|\Delta_s\|)^2 \leqslant \mathbb{E}\|\Delta_s\|^2$ to obtain
$$(\mathbb{E}\|\Delta_s(t, x^*(t))\|)^2 \;\leqslant\; \mathbb{E}\|\Delta_s(t, x^*(t))\|^2.$$

**Readable expected bound.**    Combining the above yields

$$\mathbb{E}\|e(T)\| \;\leqslant\; \left(\int_0^T e^{2\int_t^T \tilde{L}(\tau)\, d\tau}\, L_s(t)^2\, dt\right)^{1/2} \left(\int_0^T \mathbb{E}\|\Delta_s(t, x^*(t))\|^2\, dt\right)^{1/2} \tag{21}$$

**Assumption on score error.**    Using the Assumption of score error bound, which guarantees $\|\Delta_s(t, x)\| \leqslant \kappa(t)$ for all $x$, then

$$\mathbb{E}\|e(T)\| \;\leqslant\; \left(\int_0^T e^{2\int_t^T \tilde{L}(\tau)\, d\tau}\, L_s(t)^2\, dt\right)^{1/2} \left(\int_0^T \kappa(t)^2\, dt\right)^{1/2}.$$

This is exactly the equation 13 and the result of Proposition 4.2. $\qquad\square$

## D    PROOF OF COROLLARY 4.1 (GPC TIGHTENS THE TERMINAL BOUND)

**Statement restated.**    Let $\mathcal{B}(\hat{s})$ denote the upper bound on the expected sampling error derived in Proposition 4.2. If a convex combination $s_{\text{comp}} = w s_1 + (1 - w) s_2$ satisfies

$$\int_0^T \mathbb{E}\|s_{\text{comp}} - s^*\|^2 dt < \min_i \int_0^T \mathbb{E}\|s_i - s^*\|^2 dt,$$

then the corresponding theoretical error bound is strictly reduced:

$$\mathcal{B}(s_{\text{comp}}) < \min_i \mathcal{B}(s_i).$$

**Proof.**    From Proposition 4.2, the expected trajectory error for any estimator $\hat{s}$ is bounded by:

$$\mathbb{E}\|x_{\hat{s}}(T) - x^*(T)\| \;\leqslant\; \mathcal{B}(\hat{s}) := \left(\int_0^T e^{2\int_t^T \tilde{L}(\tau)\, d\tau}\, L_s(t)^2\, dt\right)^{1/2} \cdot \left(\int_0^T \mathbb{E}\|\hat{s} - s^*\|^2 dt\right)^{1/2},$$

where $\left(\int_0^T e^{2\int_t^T \tilde{L}(\tau)\, d\tau}\, L_s(t)^2\, dt\right)^{1/2} > 0$ is a system-dependent constant independent of the score estimator. The bound function $\mathcal{B}(\cdot)$ is strictly increasing with respect to the integrated mean-squared score error (MSE). Since the premise states that the integrated MSE of the composite score $s_{\text{comp}}$ is strictly smaller than that of the individual estimators $s_i$, it follows immediately that:

$$\mathcal{B}(s_{\text{comp}}) < \min_i \mathcal{B}(s_i).$$

Thus, convex score composition strictly tightens the theoretical guarantee on the trajectory simulation error. $\qquad\square$

## E    DETAILED TOOLS: NORM DERIVATIVE, INTEGRATING FACTOR, AND INEQUALITIES

### E.1    NORM DERIVATIVE INEQUALITY

Let $e : [0, T] \to \mathbb{R}^d$ be absolutely continuous. Define $\phi(t) = \|e(t)\|$. We show $\phi$ is absolutely continuous and $\phi'(t) \leqslant \|e'(t)\|$ for a.e. $t$.

**Absolute continuity.** Since $e(t) = e(0) + \int_0^t e'(\tau)\,d\tau$ with $e' \in L^1$, and the norm is 1-Lipschitz, $\phi$ is absolutely continuous.

**Difference-quotient proof.** Fix a point where $e'$ exists. Then

$$\frac{\phi(t+h) - \phi(t)}{h} = \frac{\|e(t+h)\| - \|e(t)\|}{h} \leqslant \frac{\|e(t+h) - e(t)\|}{h}.$$

Taking $h \to 0$ gives $\phi'(t) \leqslant \|e'(t)\|$. This holds for a.e. $t$.

**Chain-rule proof (when $e(t) \neq 0$).** For $g(x) = \|x\|$, $\nabla g(x) = x/\|x\|$ when $x \neq 0$. Then

$$\phi'(t) = \langle \nabla g(e(t)), e'(t) \rangle = \left\langle \frac{e(t)}{\|e(t)\|}, e'(t) \right\rangle \leqslant \|e'(t)\|.$$

At points with $e(t) = 0$, use the difference-quotient argument above.

## E.2 INTEGRATING FACTOR

Starting from $\phi'(t) \leqslant a(t)\phi(t) + b(t)$ with $\phi(0) = 0$ and $a, b \in L^1$, define $A(t) = \int_0^t a(\tau)\,d\tau$ and $g(t) = e^{-A(t)}\phi(t)$. Then

$$g'(t) = e^{-A(t)}\big(\phi'(t) - a(t)\phi(t)\big) \leqslant e^{-A(t)}b(t).$$

Integrate:

$$g(T) \leqslant \int_0^T e^{-A(t)}b(t)\,dt \quad \Rightarrow \quad \phi(T) \leqslant \int_0^T e^{A(T) - A(t)}b(t)\,dt.$$

Since $A(T) - A(t) \leqslant \int_0^T a$, a looser bound is $\phi(T) \leqslant e^{\int_0^T a}\int_0^T b(t)\,dt$.

## E.3 TONELLI, CAUCHY–SCHWARZ, AND JENSEN

Given a nonnegative integrand $H(\omega, t)$ on $\Omega \times [0, T]$, Tonelli implies

$$\mathbb{E}\left[\int_0^T H(\omega, t)\,dt\right] = \int_0^T \mathbb{E}[H(\omega, t)]\,dt.$$

For functions $f, g \in L^2([0, T])$, $\int_0^T fg \leqslant \|f\|_2\|g\|_2$. For a random variable $Z$, Jensen yields $(\mathbb{E}\|Z\|)^2 \leqslant \mathbb{E}\|Z\|^2$.

# F HOW PROPOSITIONS AND COROLLARY FIT TOGETHER

Prop. 4.1 guarantees the existence of a convex weight (often interior) that lowers the *score* MSE under mild, testable conditions (heterogeneous models reduce cross-correlation and diversify biases). Prop. 4.2 translates any reduction in (time-integrated) score MSE into a reduction of a *non-asymptotic terminal error bound*. Cor. 4.1 merely combines the two: once the functional-level inequality is strict, the certified sampling bound tightens accordingly.

## G   THE FLEXIBILITY OF GPC WITH ANY PREDICTION TYPES

A key strength of General Policy Composition (GPC) is its flexibility and independence from the specific parameterization used to train the underlying diffusion or flow-matching policies. The fundamental principle of GPC is the composition of the underlying score functions of the data distributions, $s_\theta(\tau_t, t) = \nabla_{\tau_t} \log p_t(\tau_t)$. Common parameterizations, such as noise prediction, data prediction, and v-prediction, are all mathematically inter-convertible and represent this same underlying score function. This ensures that GPC can seamlessly compose policies trained with different prediction objectives without requiring extra training.

Let's formalize the relationship between these parameterizations. The diffusion forward process defines a noisy trajectory $\tau_t$ at time $t$ from an initial trajectory $\tau_0$ and a Gaussian noise sample $\epsilon \sim \mathcal{N}(0, I)$ as:

$$\tau_t = \alpha_t \tau_0 + \sigma_t \epsilon, \tag{22}$$

where $\alpha_t$ and $\sigma_t$ are schedule-dependent coefficients.

**Score Prediction ($s$-prediction).**   This parameterization directly models the score function. The score is related to the noise $\epsilon$ by the following identity (Song et al., 2020b):

$$s(\tau_t, t) = \nabla_{\tau_t} \log p_t(\tau_t) = -\frac{\epsilon}{\sigma_t}. \tag{23}$$

Composing scores is the core of GPC. Any other parameterization can be converted to a score before composition.

**Noise Prediction ($\epsilon$-prediction).**   This is the most common parameterization, used in the original DDPM (Ho et al., 2020). The model $\epsilon_\theta(\tau_t, t)$ is trained to predict the noise $\epsilon$. A model trained on noise prediction can be converted to a score prediction model:

$$s_\theta(\tau_t, t) = -\frac{\epsilon_\theta(\tau_t, t)}{\sigma_t}. \tag{24}$$

Since the relationship is linear, composing predicted noises with weights $w_i$ is equivalent to composing the scores with the same weights.

**Data Prediction ($\tau_0$-prediction).**   This parameterization trains the model $(\tau_0)_\theta(\tau_t, t)$ to predict the original clean data $\tau_0$ from the noisy input $\tau_t$. The predicted noise $\epsilon$ can be recovered from the predicted data using the forward process definition:

$$\epsilon_\theta(\tau_t, t) = \frac{\tau_t - \alpha_t (\tau_0)_\theta(\tau_t, t)}{\sigma_t}. \tag{25}$$

This allows a data-prediction policy to be converted to the score or noise representation for composition.

**Velocity Prediction (v-prediction).**   Introduced by (Salimans & Ho, 2022), v-prediction offers improved numerical stability. The target, $\mathbf{v}$, is defined as $\mathbf{v} = \alpha_t \epsilon - \sigma_t \tau_0$. A model $\mathbf{v}_\theta(\tau_t, t)$ is trained to predict this target. We can recover the noise $\epsilon$ from a v-prediction model's output using:

$$\epsilon_\theta(\tau_t, t) = \alpha_t \mathbf{v}_\theta(\tau_t, t) + \sigma_t \tau_t. \tag{26}$$

From there, the equivalent score can be calculated.

**Implications for GPC.**   The interchangeability of these parameterizations is what makes GPC "solver-agnostic." Suppose we want to compose two policies, $\pi_1$ and $\pi_2$. If $\pi_1$ was trained using noise prediction (outputting $\epsilon_\theta^1$) and $\pi_2$ was trained using v-prediction (outputting $\mathbf{v}_\theta^2$), we can perform composition by first converting their outputs to a common representation.

For example, we can convert both to the score representation:

$$s_\theta^1(\tau_t, t) = -\frac{\epsilon_\theta^1(\tau_t, t)}{\sigma_t} \tag{27}$$

$$s_\theta^2(\tau_t, t) = -\frac{\alpha_t \mathbf{v}_\theta^2(\tau_t, t) + \sigma_t \tau_t}{\sigma_t} \tag{28}$$

Then, we can perform the convex composition in the score space:

$$s_{\text{comp}} = w_1 s_\theta^1 + w_2 s_\theta^2. \tag{29}$$

This composed score $s_{\text{comp}}$ can then be used in any standard ODE/SDE solver step to generate the next state $\tau_{t-1}$.

Alternatively, and often more direct in practice, one can convert all outputs to the noise ($\epsilon$) representation before composition, which yields an equivalent result due to the linear relationship between score and noise. This flexibility allows GPC to serve as a universal, plug-and-play module for combining a wide variety of pre-trained diffusion-based or flow-based policies, regardless of their specific training objective or parameterization.

## H    IN-DEPTH ANALYSIS ON WHAT LEADS TO THE SUCCESS OF GPC

Intuitively, GPC can be viewed as forming a product-of-experts distribution: the composed score is a convex combination of individual scores, corresponding to a (re-weighted) product of their probability densities. A higher weight on one policy simply means its density contributes more strongly to the final target distribution, concentrating probability mass on trajectories that both experts consider likely.

### H.1    THEORETICAL PERSPECTIVE: WHY CONVEX SCORE COMPOSITION CAN BE BETTER THAN INDIVIDUAL POLICIES

Based on Prop. 4.1, we show that at each diffusion timestep there exists a convex weight $w^*$ such that the composed score has a smaller error with respect to the ideal score $s^*$ than any individual policy. Prop. 4.2 extends this from a single step to the entire denoising trajectory: if at each step the composed score is closer to $s^*$, then the error along the whole trajectory accumulates more favorably, leading to a trajectory distribution closer to the ideal one. In other words, once per-step improvement in score quality is established, the stability of the generative process allows this advantage to propagate along the full trajectory, making convex score composition provably superior to relying on a single policy.

### H.2    EMPIRICAL PERSPECTIVE: WHAT DRIVES LARGER GAINS FOR SOME COMBINATIONS

At a high level, we view GPC as a way to aggregate the "good knowledge" learned by different base policies into a single, higher-likelihood target distribution, so that sampling from the composed score produces higher-quality trajectories than sampling from any individual policy alone. In practice, what counts as "good knowledge" is shaped by several factors:

**Complementary modalities provide richer information.** When GPC combines policies trained on different modalities (*e.g.*, RGB vs. point cloud), the composed score effectively leverages complementary views of the scene: RGB captures appearance, texture, and color cues, while point clouds provide precise 3D geometry and depth structure. This multimodal fusion reduces perceptual ambiguity compared to using either modality alone, and can be viewed as increasing the "informational richness" available to the composed policy. The DP+DP3 combination is a good example of this, and the visualizations of their sample distributions in Fig. 5(a) support this interpretation.

**Diverse architectures capture different inductive biases.** Even when policies share the same input modality, different architectures can encode different inductive biases and error patterns. For instance, $\pi 0$ is DiT-based, whereas Flow Policy uses a U-Net-style backbone. They may model similar underlying action distributions but emphasize different structures in the data. GPC can exploit this diversity by aggregating their strengths while averaging out idiosyncratic weaknesses, which we see reflected in improved performance and the qualitative analysis in Fig. 5(b).

**Task-specific strengths shape composition benefits.** Task choice also matters: in some tasks, one base policy may be significantly weaker. In such cases, naive averaging (or an unbalanced weight) can let the weaker policy drag down the overall performance. Our analysis in Sec. 6.3 (Finding 3) shows that GPC performs best when the *better-performing base policy receives a larger weight*, which is especially important when task difficulty or mismatch affects one policy more than the other. Thus, task-specific policy strengths and the ability to adjust weights are key factors behind the observed gains.

**Weight selection matters in the composed distribution.** Our theory guarantees the existence of an optimal weight $w^*$, but in practice GPC still requires choosing concrete weights. These weights control how much each base distribution contributes to the final composed distribution and therefore directly influence how much probability mass is placed on high-quality trajectories. This is why we performed extensive weight-sweep studies: to show how different weights affect performance, and to provide empirical guidance on weight selection for real deployments. We view designing better, possibly adaptive, strategies to approximate $w^*$ as an important direction for future work.

# I EXPERIMENT DETAILS

## I.1 ROBOMIMIC

The Robomimic benchmark (Mandlekar et al., 2022) includes three manipulation tasks: Can, Lift, and Square. We train all baselines with batch size 1024 for 1000 epochs. Training uses DDIM sampling with the scaled linear beta scheduler and prediction with epsilon. Diffusion steps are set to 100 during training and 10 at inference. Each model is trained with observation horizon = 2 and chunk size = 16. Evaluation is performed across 20 parallel environments, each running 10 episodes, giving a total of 200 rollouts. The original code of Robomimic is from `https://github.com/ARISE-Initiative/robomimic`. We reproduce the baselines based on the codes from `https://github.com/EDiRobotics/mimictest`.

## I.2 PUSHT

The PushT benchmark (Florence et al., 2021) involves planar pushing in a 2D workspace. Here, training uses batch size 256 and runs for 500 epochs, with all other parameters kept identical to Robomimic. Evaluation follows the same protocol of 200 rollouts. The original code of Robomimic is from `https://github.com/real-stanford/diffusion_policy` and `https://github.com/google-research/ibc`. We reproduce the baselines based on the codes from `https://github.com/EDiRobotics/mimictest`.

## I.3 ROBOTWIN

RoboTwin (Mu et al., 2025) is a dual-arm manipulation benchmark that combines real-world teleoperated demonstrations with high-fidelity synthetic data, offering a standardized platform for studying large-scale manipulation learning. The extended RoboTwin 2.0 (Chen et al., 2025) release covers more than 50 tasks, supporting diverse and complex scenarios. Baselines are reproduced based on the codes from `https://github.com/RoboTwin-Platform/RoboTwin/tree/RoboTwin-1.0` and `https://github.com/RoboTwin-Platform/RoboTwin/tree/main`. The success rate of each task is determined with 100 rollouts.

For our experiments, we evaluate on a curated subset of tasks:

- *RoboTwin 1.0:* Empty Cup Place, Dual Bottles Pick (Hard), Dual Bottles Pick (Easy), Shoe Place, Dual Shoes Place, Pick Apple Messy, Block Hammer Beat.
- *RoboTwin 2.0:* Hanging Mug, Open Laptop, Place Burger Fries, Put Object Cabinet, Stack Bowls, Three Turn Switch.

The $DP_{img}$ and $DP_{pcd}$ correspond to the diffusion policy based on RGB images (*i.e.*, DP (Chi et al., 2023)) and point cloud (*i.e.*, DP3 (Ze et al., 2024b)), respectively. In RoboTwin 1.0, we reproduce the $DP_{img}$ and $DP_{pcd}$ (without using point cloud color) with random seed 0. Since the diffusion scores from different policies are composed at each denoising step (Alg. 1), we unify the training settings of both $DP_{pcd}$ and $DP_{img}$. In particular, they are trained with DDPM with 100 training and inference steps. In RoboTwin 2.0 experiments, we train DPs with the same settings as RDT to ensure compatibility so that our GPC can be applied consistently. For example, RDT employs sample prediction, we align our diffusion models accordingly by training both $DP_{img}$ and $DP_{pcd}$ under the same prediction setting.

**Task Prompt in RoboTwin.** Here we present the detailed text description and its corresponding text prompt for VLAs. For example, for the Place Burger task, the task description and schema are:

- **Full description:** "Use dual arm to pick the hamburg and frenchfries and put them onto the tray."
- **Schema:** "A denotes the hamburg, B denotes the tray, C denotes the frenchfries"

During training, the VLA receives diverse paraphrased prompts for this task, such as:

- "Use both arms to move A and C to B."

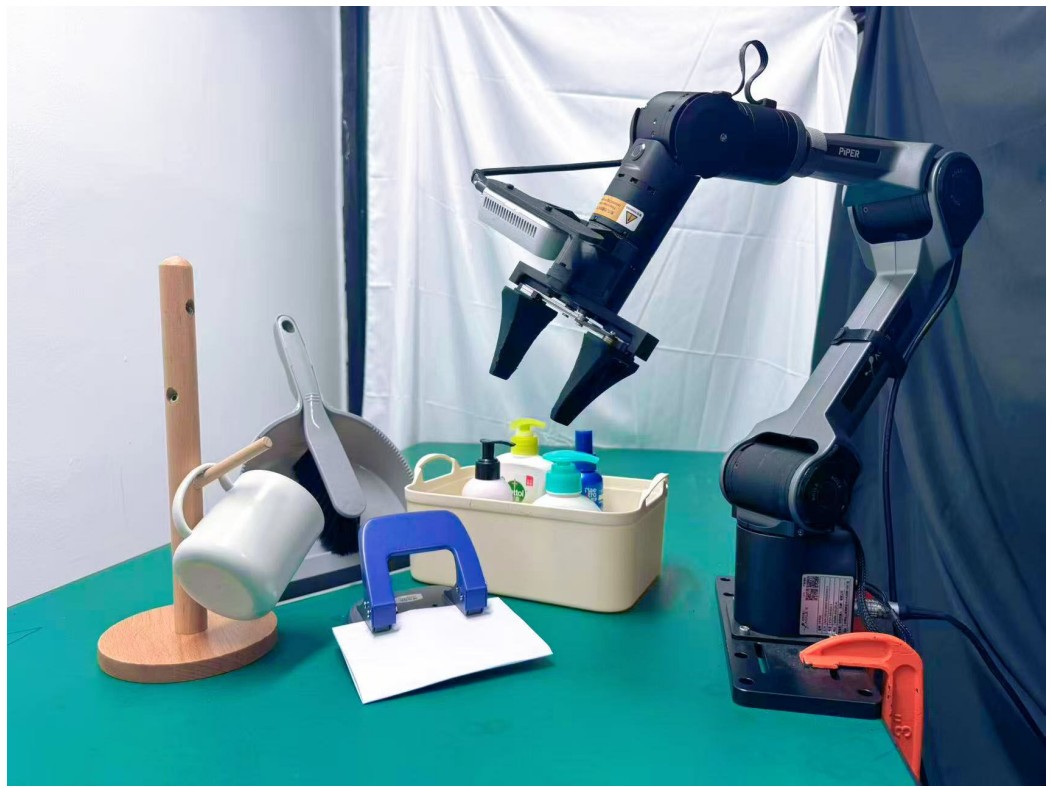

Figure 7: **Illustration of Experimental Setup.**

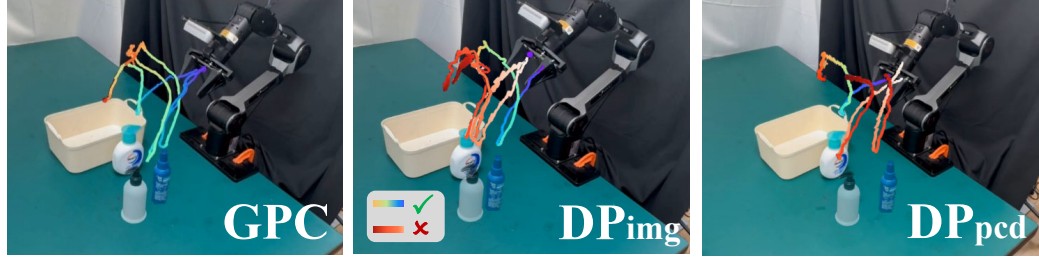

Figure 8: **Tracking Results of Real-world Experiment in Place Bottles.**

- "Lift A and C, placing them neatly on B"

At test time, we use unseen paraphrases with the same schema, *e.g.*.:

- "Pick A and C, then place them on B."
- "Grab A and C together, setting them on B."

## I.4 REAL-WORLD EXPERIMENTS

We choose DP$_{img}$ and DP$_{pcd}$ as our base policies for real-world experiments. For DP$_{img}$, we use an Intel RealSense D435 RGB camera at $640 \times 480$ resolution (primary view and wrist view) to get the RGB images. For DP$_{pcd}$, we use an Intel RealSense L515 depth camera at $640 \times 480$, where we obtain point clouds by using depth images together with camera intrinsics. The robot platform is *Piper*, operated in a master–slave teleoperation setup. The illustration of the real-world experimental setup is shown in Fig. 7. Our GPC achieves superior performance compared with the base policies,

presenting better trajectories in Fig. 8. Training follows official configurations: $DP_{pcd}$ is trained for 600 epochs with batch size 256 (official code), while $DP_{img}$ is trained for 20k steps with batch size 64 (Lerobot (Cadene et al., 2024) diffusion implementation).

### I.5 NOTATION FOR GPC FLEXIBILITY

Notably, the prediction types in diffusion models are not strictly restricted to a single formulation (*e.g.*, $\epsilon$-prediction, $x_0$-prediction, or $v$-prediction), but can be freely combined within our framework. When heterogeneous prediction types are adopted simultaneously, the denoising process requires proper alignment to ensure consistency. We provide detailed guidance in Sec. G on how to reconcile different prediction types in GPC from a theoretical perspective, further demonstrating the flexibility of our proposed method.

## J ADDITIONAL EXPERIMENTAL RESULTS

In this section, we provide the complete set of experimental results to complement the main text. These results include all weight configurations for convex score composition, as well as the outcomes under logical AND and OR operators.

**Robomimic and PushT.** We report detailed results on the Robomimic (Can, Lift, Square) and PushT tasks. In addition to the average performance reported in the main paper, we include (i) the full tables for convex score combination, logical AND, and logical OR composition in Tab. 10, and (ii) the breakdown of performance under different convex weights $w$ for each task: Can (Tab. 11 & Fig. 9), Square (Tab. 13 & Fig. 10), Lift (Tab. 12 & Fig. 11) and PushT (Tab. 14 & Fig. 12). These results illustrate how GPC adapts across weighting configurations and provide insight into the trade-offs between modalities and model backbones.

**RoboTwin.** We also provide the complete results on RoboTwin 2.0 across all tasks. In particular, we include full tables comparing base policies and their compositions (*e.g.*, $DP_{img}$ + $DP_{pcd}$, RDT + $DP_{pcd}$), with all tested weight settings: Open Labtop (Tab. 15), Place Burger (Tab. 16 & Fig. 21), Put Object Cabinet (Tab. 17 & Fig. 24), Hanging Mug (Tab. 18 & Fig. 23), Stack Bowls Three (Tab. 19 & Fig. 14) and Turn Switch (Tab. 20 & Fig. 20). These detailed numbers confirm the robustness of GPC across diverse manipulation tasks and further validate the findings in Sec. 6.

**Real-world Experiments.** We further report complete results for the four real-world tasks: Place Bottles (Tab. 21 & Fig. 27), Hang Mug (Tab. 22 & Fig. 26), Clean Table (Tab. 23 & Fig. 25), and Punch Holes (Tab. 24 & Fig. 28). Similar to the simulation benchmarks, we provide full comparisons between base policies and their GPC compositions under all tested weight settings. These results consistently show that GPC achieves higher success rates than individual policies, thereby confirming its effectiveness in practical robotic scenarios.

**Summary.** Together, these extended results give a comprehensive view of GPC's empirical behavior across different operators and weightings. They serve as a reference for understanding not only the average improvements but also the sensitivity of performance to the choice of weights and composition strategies.

Table 10: **Experiments on Robomimic and PushT with GPC under convex score combination, Logical AND and Logical OR.**

| Method | Generative Mode | Model Type | Robomimic | | | PushT | Average |
|---|---|---|---|---|---|---|---|
| | | | Can | Lift | Square | PushT | |
| *Base Policies* | | | | | | | |
| Diffusion Policy (DP) | Diffusion | VA | 34.50 | 98.50 | 2.00 | 21.75 | 39.19 |
| Mamba Policy (MP) | Flow Matching | VA | 5.00 | 98.50 | 3.00 | 12.06 | 29.64 |
| Flow Policy (FP) | Diffusion | VA | 95.00 | 13.00 | 77.50 | 54.25 | 59.94 |
| Florence Policy-D | Diffusion | VLA | 61.50 | 97.00 | 46.50 | 40.00 | 61.25 |
| Florence Policy-F | Flow Matching | VLA | 89.00 | 98.50 | 88.50 | 39.38 | 78.84 |
| $\pi 0$ | Flow Matching | VLA | 96.50 | 99.00 | 92.50 | 57.69 | 86.42 |
| *Composed Policies via Convex Score Combination* | | | | | | | |
| DP+MP | Diffusion | VA & VA | 34.50 | 99.50 | 8.00 | 23.63 | 41.41 +2.22 |
| Florence-Policy-D+DP | Diffusion | VLA & VA | 62.50 | 100.00 | 61.50 | 43.06 | 66.76 +5.51 |
| Florence-Policy-D+MP | Diffusion | VLA & VA | 63.00 | 100.00 | 54.50 | 40.88 | 64.60 +3.35 |
| Florence-Policy-F+FP | Flow Matching | VLA & VA | 98.50 | 98.50 | 92.50 | 56.06 | 86.39 +7.55 |
| $\pi 0$+FP | Flow Matching | VLA & VA | 99.50 | 100.00 | 94.00 | 62.25 | 88.94 +2.52 |
| *Composed Policies via Logical AND Composition* | | | | | | | |
| DP+MP | Diffusion | VA & VA | 84.00 | 99.50 | 48.00 | 28.18 | 64.92 +25.73 |
| Florence-Policy-D+DP | Diffusion | VLA & VA | 90.50 | 100.00 | 90.00 | 36.31 | 79.20 +17.95 |
| Florence-Policy-D+MP | Diffusion | VLA & VA | 83.00 | 100.00 | 90.00 | 37.38 | 77.60 +16.35 |
| *Composed Policies via Logical OR Composition* | | | | | | | |
| DP+MP | Diffusion | VA & VA | 82.50 | 99.50 | 44.00 | 29.13 | 63.78 +24.59 |
| Florence-Policy-D+DP | Diffusion | VLA & VA | 83.50 | 100.00 | 89.00 | 37.87 | 77.59 +16.34 |
| Florence-Policy-D+MP | Diffusion | VLA & VA | 86.50 | 100.00 | 86.50 | 38.44 | 77.86 +16.61 |

Table 11: **Experiments on Robomimic Can with GPC under different weighting.**

| Method | Generative Mode | Model Type | Can | | | | | | | | | | |
|---|---|---|---|---|---|---|---|---|---|---|---|---|---|
| | | | 0.0 | 0.1 | 0.2 | 0.3 | 0.4 | 0.5 | 0.6 | 0.7 | 0.8 | 0.9 | 1.0 |
| DP+Mamba Policy | Diffusion | VA & VA | 5.00 | 10.00 | 10.50 | 10.50 | 16.50 | 20.00 | 20.00 | 23.00 | 25.00 | 29.00 | **34.50** |
| Florence DiT + DP | Diffusion | VLA & VA | 34.50 | 34.50 | 42.50 | 48.00 | 56.00 | 62.50 | 60.50 | **63.50** | 58.00 | 62.50 | 61.50 |
| Florence DiT + MambaP | Diffusion | VLA & VA | 5.00 | 11.50 | 21.50 | 30.50 | 39.00 | 44.50 | 47.50 | 46.50 | 56.50 | **63.00** | 61.50 |
| Florence Flow+FlowP | Flow Matching | VLA & VA | 95.00 | **98.50** | **98.50** | 96.00 | 96.50 | 97.00 | 93.00 | 92.00 | 90.00 | 90.50 | 89.00 |
| $\pi 0$+FlowP | Flow Matching | VLA & VA | 95.00 | 96.00 | **99.00** | 98.00 | 97.50 | 98.50 | 99.50 | 98.00 | 96.00 | 96.00 | 96.50 |

Table 12: **Experiments on Robomimic Lift with GPC under different weighting.**

| Method | Generative Mode | Model Type | Lift | | | | | | | | | | |
|---|---|---|---|---|---|---|---|---|---|---|---|---|---|
| | | | 0.0 | 0.1 | 0.2 | 0.3 | 0.4 | 0.5 | 0.6 | 0.7 | 0.8 | 0.9 | 1.0 |
| DP+Mamba Policy | Diffusion | VA & VA | 98.50 | 99.00 | **99.50** | 96.50 | 99.00 | 98.50 | 98.50 | 98.50 | 98.50 | 98.50 | 98.50 |
| Florence DiT + DP | Diffusion | VLA & VA | 98.50 | 99.50 | 99.00 | **100.00** | 99.50 | 99.50 | 99.50 | 99.50 | 99.50 | 98.00 | 97.00 |
| Florence DiT + MambaP | Diffusion | VLA & VA | 98.50 | **100.00** | 99.50 | 99.00 | 99.50 | 99.00 | 99.00 | 97.50 | 98.50 | 97.00 | 97.00 |
| Florence Flow+FlowP | Flow Matching | VLA & VA | 13.00 | 10.50 | 12.50 | 23.50 | 55.00 | 81.50 | 93.00 | 98.00 | **100.00** | 98.50 | 98.50 |
| $\pi 0$+FlowP | Flow Matching | VLA & VA | 13.00 | 12.50 | 17.00 | 32.50 | 67.50 | 92.50 | 98.50 | **100.00** | **100.00** | 99.50 | 99.00 |

Table 13: **Experiments on Robomimic Square with GPC under different weighting.**

| Method | Generative Mode | Model Type | Square | | | | | | | | | | |
|---|---|---|---|---|---|---|---|---|---|---|---|---|---|
| | | | 0.0 | 0.1 | 0.2 | 0.3 | 0.4 | 0.5 | 0.6 | 0.7 | 0.8 | 0.9 | 1.0 |
| DP+Mamba Policy | Diffusion | VA & VA | 3.00 | 3.00 | 4.00 | 1.50 | **8.00** | 4.50 | 6.00 | 6.00 | 3.50 | 7.50 | 2.00 |
| Florence DiT + DP | Diffusion | VLA & VA | 2.00 | 12.50 | 20.00 | 34.00 | 44.00 | 49.00 | **61.50** | 57.00 | 59.50 | 54.50 | 46.50 |
| Florence DiT + MambaP | Diffusion | VLA & VA | 3.00 | 8.00 | 8.50 | 17.00 | 22.00 | 34.00 | 45.00 | 45.50 | 50.00 | **54.50** | 46.50 |
| Florence Flow+FlowP | Flow Matching | VLA & VA | 77.50 | 79.00 | 85.00 | 92.00 | 92.00 | 92.00 | 91.00 | 88.00 | 88.50 | **92.50** | 88.50 |
| $\pi 0$+FlowP | Flow Matching | VLA & VA | 77.50 | 80.50 | 84.50 | **94.00** | 93.50 | **94.00** | 93.50 | 93.00 | 90.50 | 93.50 | 92.50 |

Table 14: **Experiments on PushT with GPC under different weighting.**

| Method | Generative Mode | Model Type | PushT | | | | | | | | | | |
|---|---|---|---|---|---|---|---|---|---|---|---|---|---|
| | | | 0.0 | 0.1 | 0.2 | 0.3 | 0.4 | 0.5 | 0.6 | 0.7 | 0.8 | 0.9 | 1.0 |
| DP+Mamba Policy | Diffusion | VA & VA | 12.06 | 19.81 | 18.31 | 18.87 | 19.94 | 19.88 | 18.13 | 21.50 | **23.63** | 22.38 | 21.75 |
| Florence DiT + DP | Diffusion | VLA & VA | 21.75 | 26.75 | 29.38 | 32.75 | 36.06 | 39.69 | 41.13 | **43.06** | 40.50 | 40.56 | 40.00 |
| Florence DiT + MambaP | Diffusion | VLA & VA | 12.06 | 22.88 | 25.81 | 30.62 | 33.94 | 37.00 | 38.44 | 40.50 | 40.75 | **40.88** | 40.00 |
| Florence Flow+FlowP | Flow Matching | VLA & VA | 54.25 | **56.06** | 54.50 | 50.81 | 47.38 | 48.31 | 47.69 | 50.50 | 46.19 | 40.75 | 39.38 |
| $\pi$0+FlowP | Flow Matching | VLA & VA | 54.25 | 54.31 | 56.81 | 56.37 | 53.31 | 57.69 | 59.12 | 61.50 | **62.25** | 61.50 | 57.69 |

Table 15: **Experiments on RoboTwin Open Laptop with GPC under different weighting.**

| Method | Generative Mode | Model Type | RoboTwin: Open Laptop | | | | | | | | | | |
|---|---|---|---|---|---|---|---|---|---|---|---|---|---|
| | | | 0.0 | 0.1 | 0.2 | 0.3 | 0.4 | 0.5 | 0.6 | 0.7 | 0.8 | 0.9 | 1.0 |
| DP+DP3 | Diffusion | VA & VA | **0.93** | **0.93** | 0.92 | **0.93** | **0.93** | 0.87 | 0.84 | 0.79 | 0.77 | 0.74 | 0.74 |
| RDT + DP | Diffusion | VLA & VA | 0.74 | 0.74 | 0.77 | 0.78 | 0.79 | **0.80** | 0.75 | 0.76 | 0.73 | 0.68 | 0.69 |
| RDT + DP3 | Diffusion | VLA & VA | 0.93 | 0.92 | 0.92 | 0.91 | 0.92 | **0.94** | 0.91 | 0.86 | 0.77 | 0.67 | 0.69 |

Table 16: **Experiments on RoboTwin Place Burger Fries with GPC under different weighting.**

| Method | Generative Mode | Model Type | RoboTwin: Place Burger Fries | | | | | | | | | | |
|---|---|---|---|---|---|---|---|---|---|---|---|---|---|
| | | | 0.0 | 0.1 | 0.2 | 0.3 | 0.4 | 0.5 | 0.6 | 0.7 | 0.8 | 0.9 | 1.0 |
| DP+DP3 | Diffusion | VA & VA | 0.72 | 0.73 | **0.78** | 0.74 | 0.72 | 0.65 | 0.68 | 0.64 | 0.66 | 0.54 | 0.49 |
| RDT + DP | Diffusion | VLA & VA | 0.49 | 0.54 | 0.56 | **0.57** | 0.53 | 0.49 | 0.50 | 0.48 | 0.45 | 0.45 | 0.46 |
| RDT + DP3 | Diffusion | VLA & VA | 0.72 | 0.79 | **0.83** | **0.83** | 0.77 | 0.78 | 0.72 | 0.67 | 0.67 | 0.55 | 0.46 |

Table 17: **Experiments on RoboTwin Put Object Cabinet with GPC under different weighting.**

| Method | Generative Mode | Model Type | RoboTwin: Put Object Cabinet | | | | | | | | | | |
|---|---|---|---|---|---|---|---|---|---|---|---|---|---|
| | | | 0.0 | 0.1 | 0.2 | 0.3 | 0.4 | 0.5 | 0.6 | 0.7 | 0.8 | 0.9 | 1.0 |
| DP+DP3 | Diffusion | VA & VA | 0.71 | 0.80 | **0.82** | 0.73 | 0.66 | 0.73 | 0.63 | 0.67 | 0.67 | 0.55 | 0.56 |
| RDT + DP | Diffusion | VLA & VA | 0.56 | 0.54 | **0.59** | 0.54 | 0.51 | 0.52 | 0.40 | 0.35 | 0.27 | 0.32 | 0.32 |
| RDT + DP3 | Diffusion | VLA & VA | 0.71 | 0.71 | **0.78** | 0.71 | 0.69 | 0.61 | 0.58 | 0.46 | 0.37 | 0.40 | 0.32 |

Table 18: **Experiments on RoboTwin Hanging Mug with GPC under different weighting.**

| Method | Generative Mode | Model Type | RoboTwin: Hanging Mug | | | | | | | | | | |
|---|---|---|---|---|---|---|---|---|---|---|---|---|---|
| | | | 0.0 | 0.1 | 0.2 | 0.3 | 0.4 | 0.5 | 0.6 | 0.7 | 0.8 | 0.9 | 1.0 |
| DP+DP3 | Diffusion | VA & VA | 0.21 | **0.23** | 0.20 | 0.22 | 0.18 | 0.17 | 0.16 | 0.11 | 0.15 | 0.11 | 0.10 |
| RDT + DP | Diffusion | VLA & VA | 0.10 | 0.13 | 0.09 | 0.15 | 0.11 | **0.18** | **0.18** | 0.15 | 0.14 | 0.12 | 0.13 |
| RDT + DP3 | Diffusion | VLA & VA | 0.21 | 0.26 | 0.31 | 0.30 | **0.36** | 0.25 | 0.25 | 0.22 | 0.24 | 0.15 | 0.13 |

Table 19: **Experiments on RoboTwin Stack Bowls Three with GPC under different weighting.**

| Method | Generative Mode | Model Type | RoboTwin: Stack Bowls Three | | | | | | | | | | |
|---|---|---|---|---|---|---|---|---|---|---|---|---|---|
| | | | 0.0 | 0.1 | 0.2 | 0.3 | 0.4 | 0.5 | 0.6 | 0.7 | 0.8 | 0.9 | 1.0 |
| DP+DP3 | Diffusion | VA & VA | 0.64 | 0.70 | 0.66 | **0.71** | 0.60 | 0.53 | 0.63 | 0.56 | 0.59 | 0.49 | 0.52 |
| RDT + DP | Diffusion | VLA & VA | 0.52 | 0.65 | **0.66** | 0.57 | 0.66 | 0.59 | 0.58 | 0.50 | 0.40 | 0.32 | 0.47 |
| RDT + DP3 | Diffusion | VLA & VA | 0.64 | 0.71 | **0.73** | 0.55 | 0.71 | 0.70 | 0.60 | 0.59 | 0.48 | 0.42 | 0.47 |

Table 20: **Experiments on RoboTwin Turn Switch with GPC under different weighting.**

| Method | Generative Mode | Model Type | RoboTwin: Turn Switch | | | | | | | | | | |
|---|---|---|---|---|---|---|---|---|---|---|---|---|---|
| | | | 0.0 | 0.1 | 0.2 | 0.3 | 0.4 | 0.5 | 0.6 | 0.7 | 0.8 | 0.9 | 1.0 |
| DP+DP3 | Diffusion | VA & VA | **0.71** | 0.68 | 0.60 | 0.63 | 0.67 | 0.56 | 0.50 | 0.45 | 0.41 | 0.42 | 0.38 |
| RDT + DP | Diffusion | VLA & VA | **0.38** | 0.28 | 0.31 | 0.28 | 0.36 | 0.37 | 0.34 | 0.30 | **0.38** | 0.35 | 0.30 |
| RDT + DP3 | Diffusion | VLA & VA | **0.71** | 0.52 | 0.54 | 0.48 | 0.51 | 0.59 | 0.51 | 0.43 | 0.42 | 0.45 | 0.30 |

Table 21: **Experiments on Real-world Place Bottle with GPC under different weighting.**

| Method | Generative Mode | Model Type | Real-world: Place Bottle | | | | | |
|---|---|---|---|---|---|---|---|---|
| | | | 0.0 | 0.2 | 0.4 | 0.6 | 0.8 | 1.0 |
| DP+DP3 | Diffusion | VA & VA | 11/20 | **13/20** | 11/20 | 12/20 | 10/20 | 7/20 |

Table 22: **Experiments on Real-world Hang Mug with GPC under different weighting.**

| Method | Generative Mode | Model Type | Real-world: Hang Mug | | | | | |
|---|---|---|---|---|---|---|---|---|
| | | | 0.0 | 0.2 | 0.4 | 0.6 | 0.8 | 1.0 |
| DP+DP3 | Diffusion | VA & VA | 6/20 | **7/20** | 5/20 | **7/20** | 6/20 | 5/20 |

Table 23: **Experiments on Real-world Clean Table with GPC under different weighting.**

| Method | Generative Mode | Model Type | Real-world: Clean Table | | | | | |
|---|---|---|---|---|---|---|---|---|
| | | | 0.0 | 0.2 | 0.4 | 0.6 | 0.8 | 1.0 |
| DP+DP3 | Diffusion | VA & VA | 7/20 | 7/20 | **14/20** | 10/20 | 12/20 | 12/20 |

Table 24: **Experiments on Real-world Punch Holes with GPC under different weighting.**

| Method | Generative Mode | Model Type | Real-world: Punch Holes | | | | | |
|---|---|---|---|---|---|---|---|---|
| | | | 0.0 | 0.2 | 0.4 | 0.6 | 0.8 | 1.0 |
| DP+DP3 | Diffusion | VA & VA | 6/20 | 6/20 | 5/20 | 7/20 | **9/20** | 7/20 |

## K  VISUALIZATION ON ROBOT TASKS

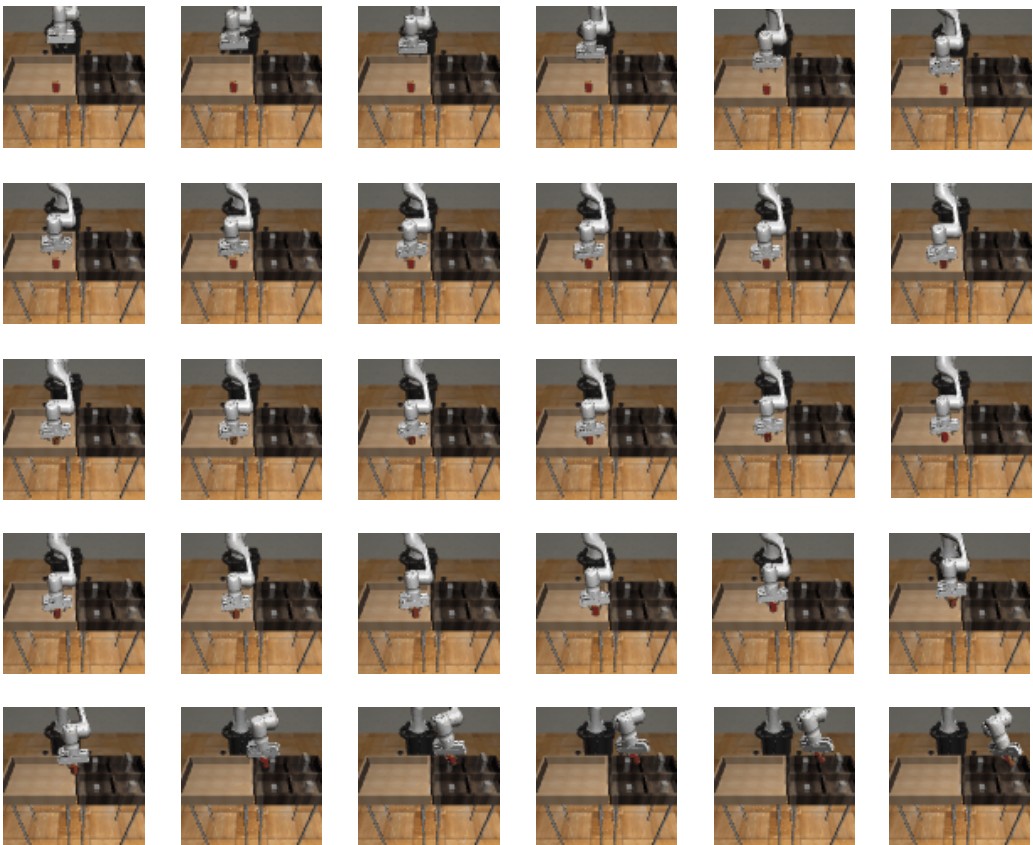

Figure 9: **Illustration of Robomimic Can.**

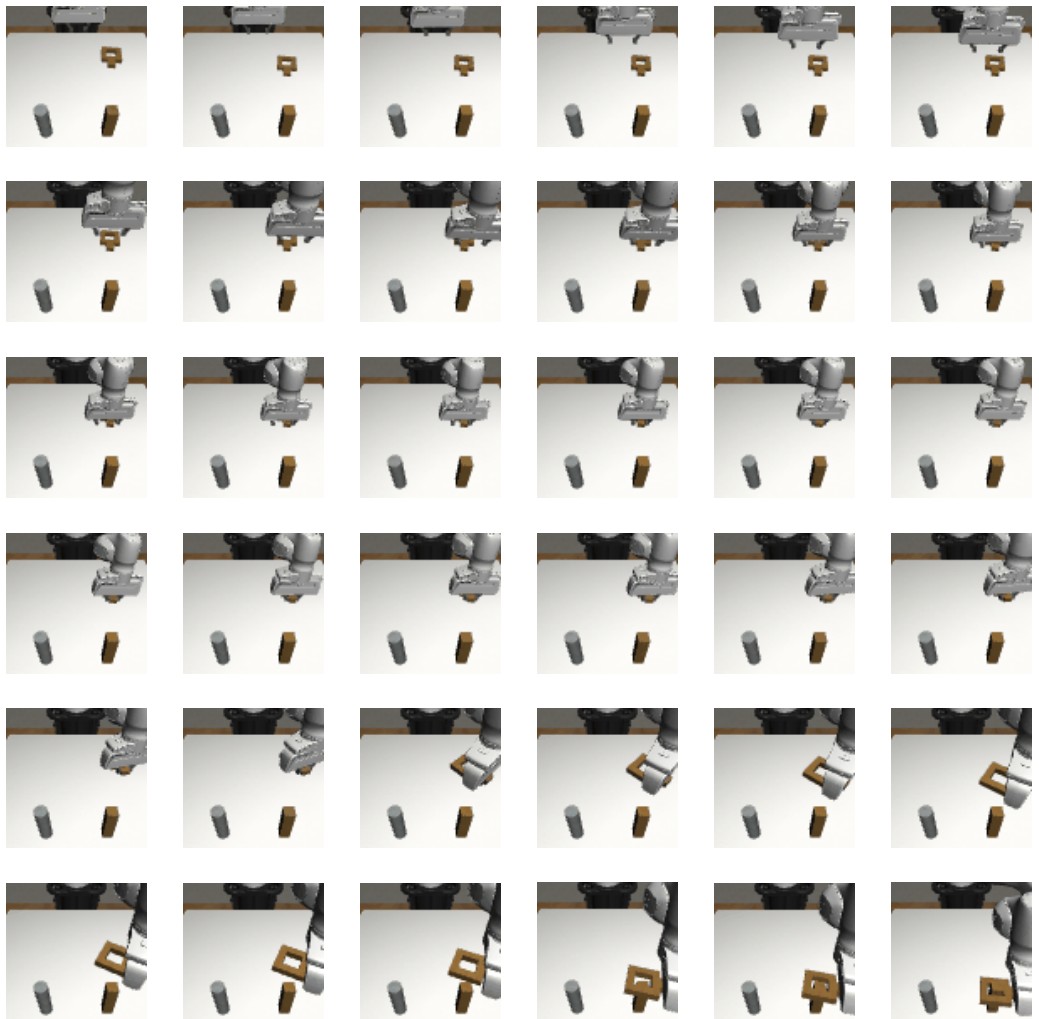

Figure 10: **Illustration of Robomimic Square.**

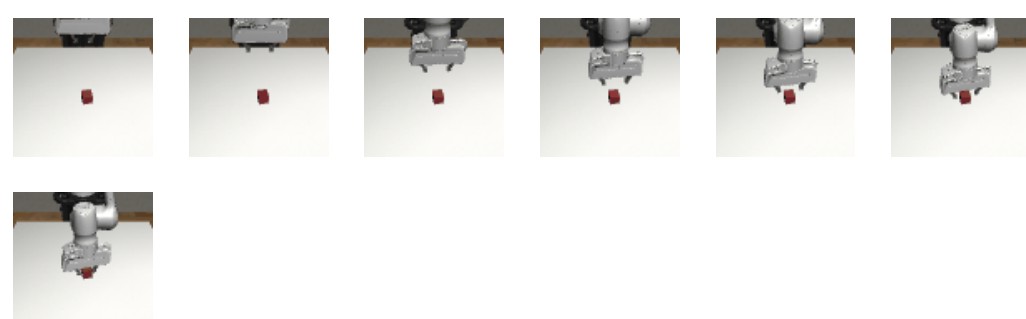

Figure 11: **Illustration of Robomimic Lift.**

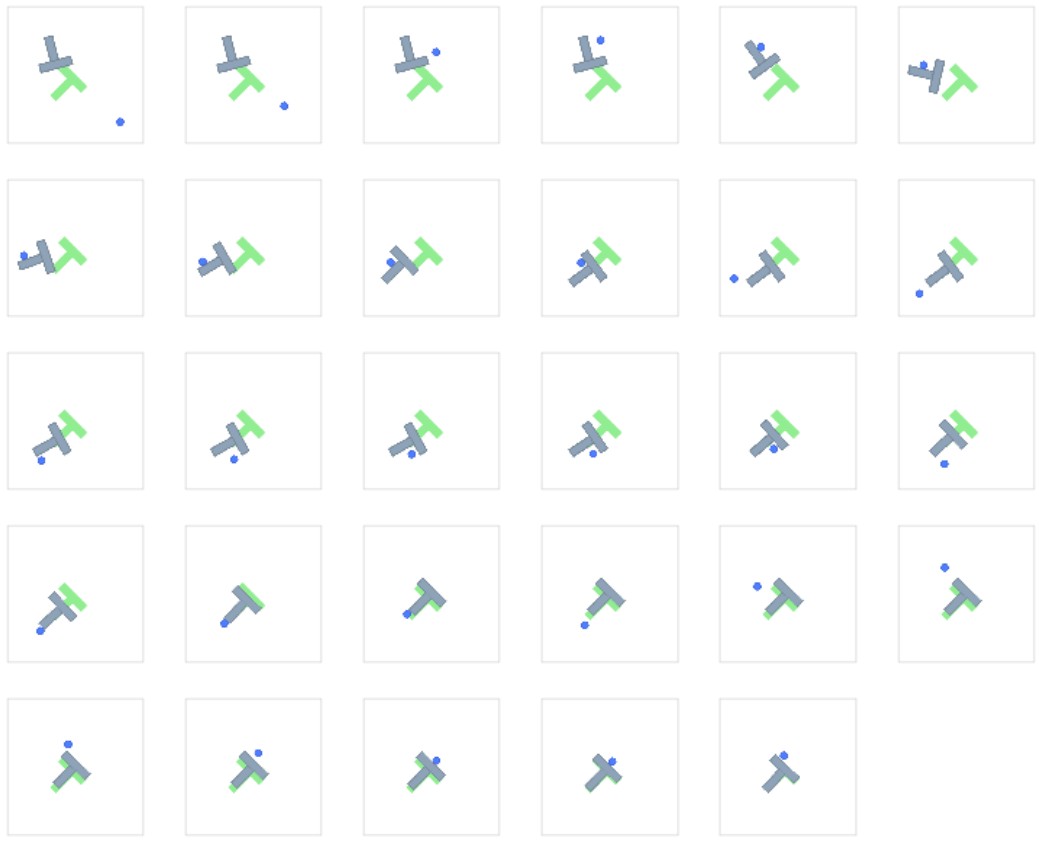

Figure 12: **Illustration of PushT.**

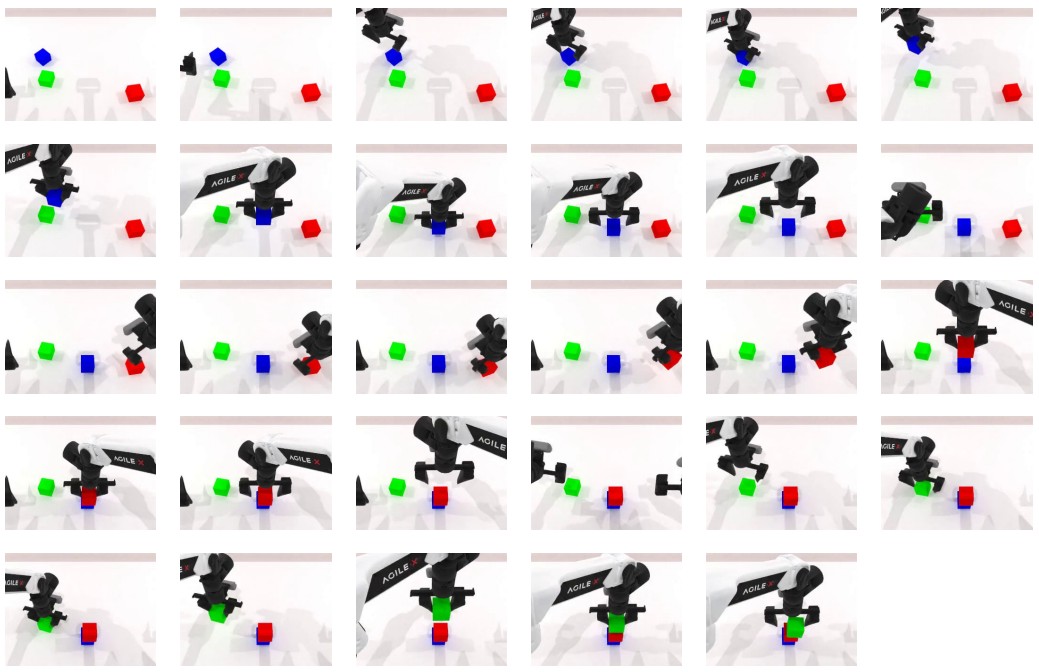

Figure 13: **Illustration of RoboTwin 1.0 Blocks Stack (Hard).**

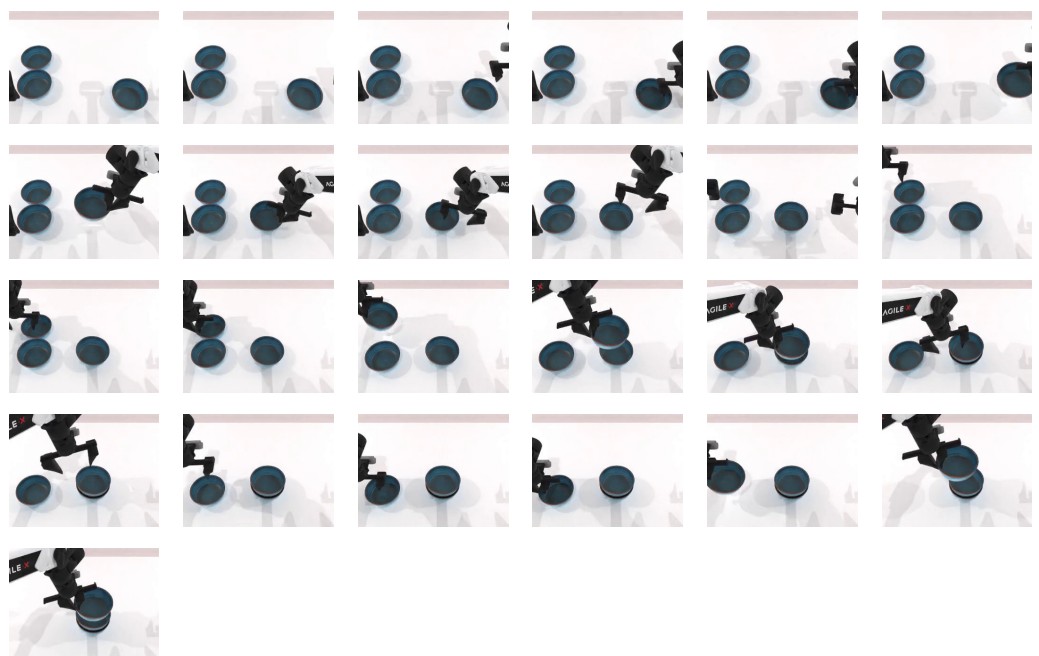

Figure 14: **Illustration of RoboTwin 1.0 Bowl Stack.**

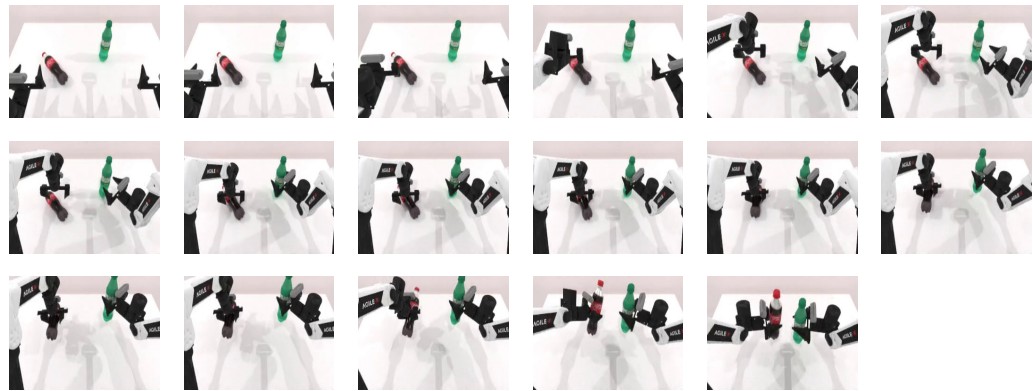

Figure 15: **Illustration of RoboTwin 1.0 Dual Bottle Pick Hard.**

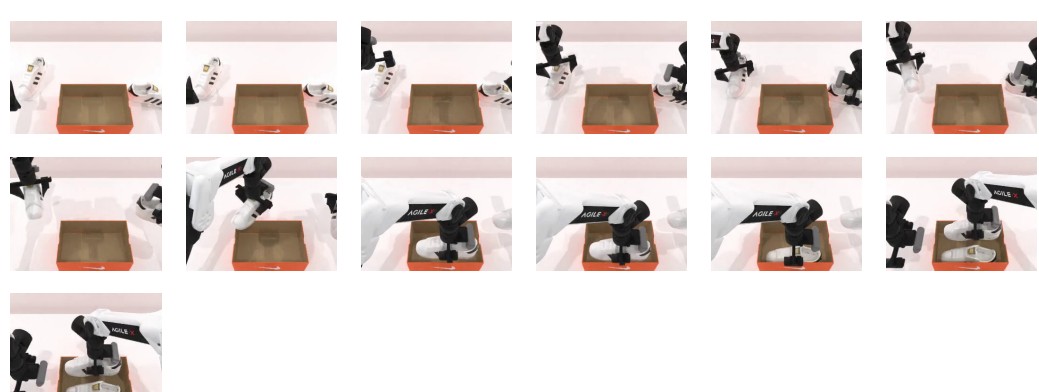

Figure 16: **Illustration of RoboTwin 1.0 Dual Shoes Place.**

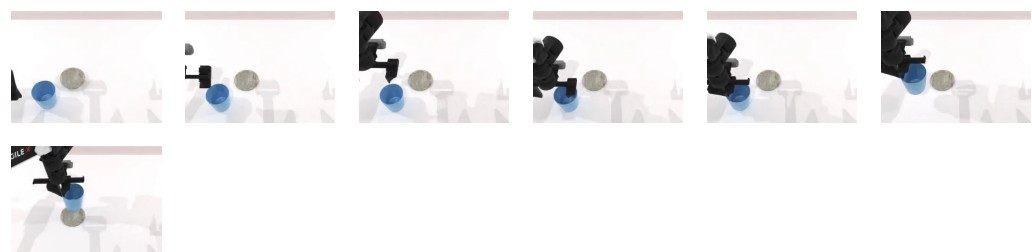

Figure 17: **Illustration of RoboTwin 1.0 Empty Cup Place.**

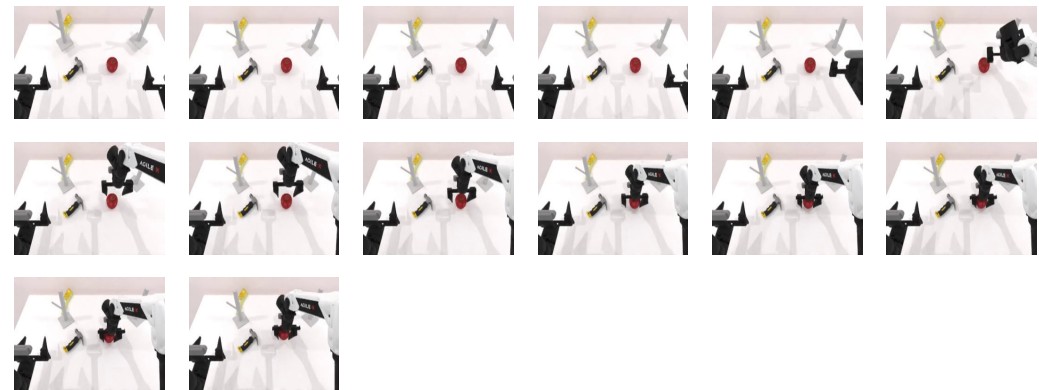

Figure 18: **Illustration of RoboTwin 1.0 Pick Apple Messy.**

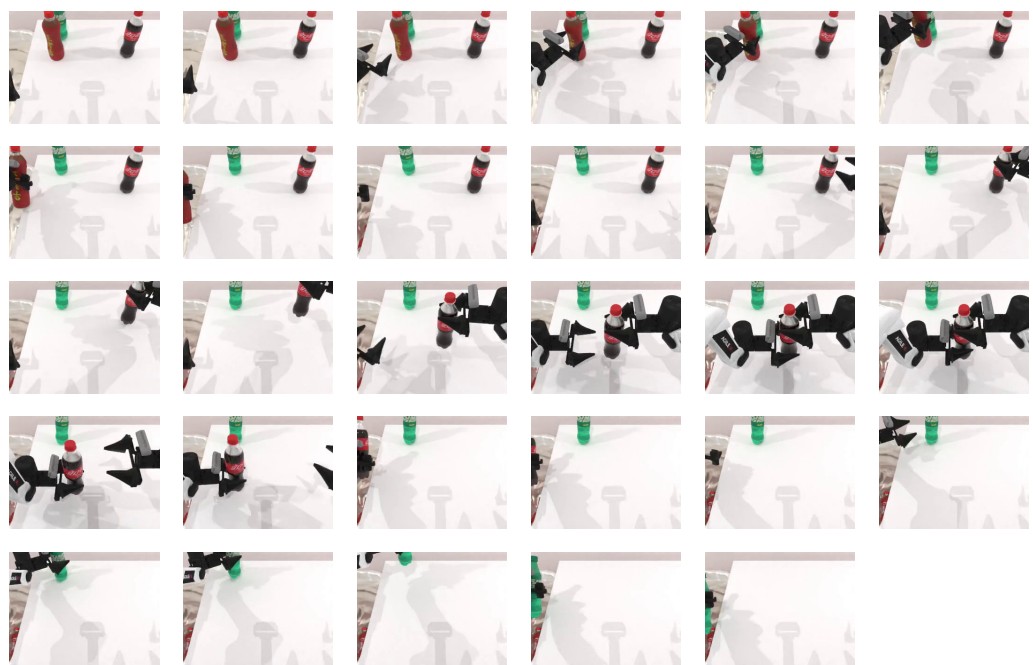

Figure 19: **Illustration of RoboTwin 1.0 Put Bottles Dustbin.**

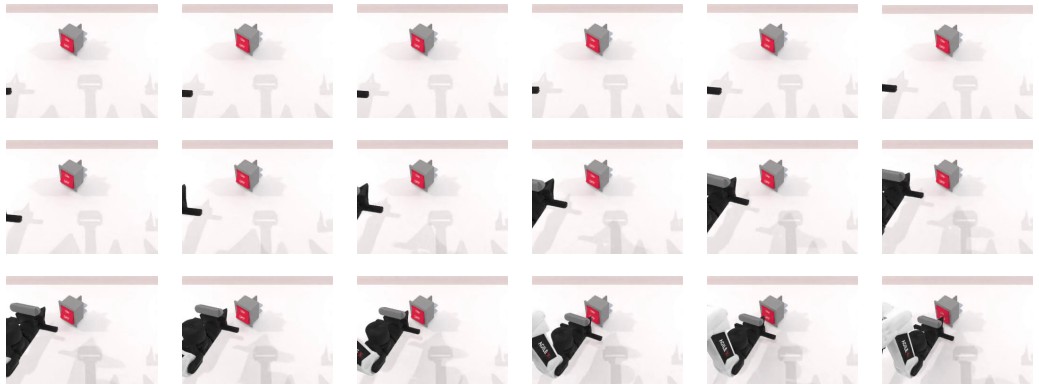

Figure 20: **Illustration of RoboTwin 2.0 Turn Switch.**

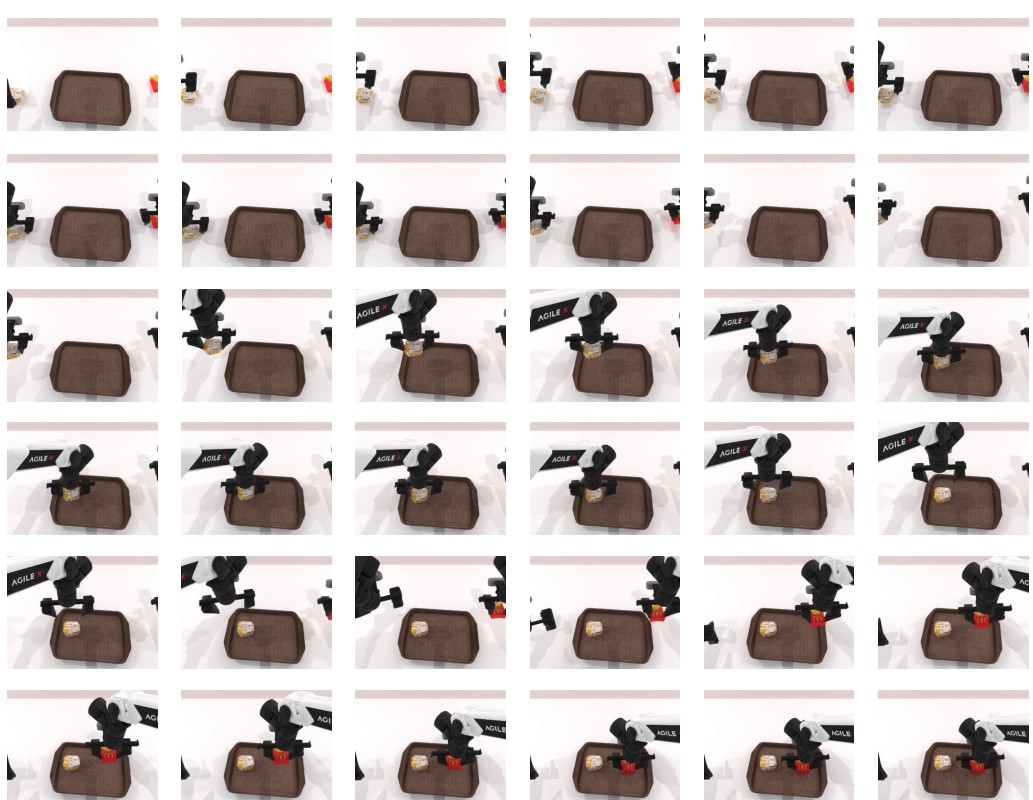

Figure 21: **Illustration of RoboTwin 2.0 Place Burger Fries.**

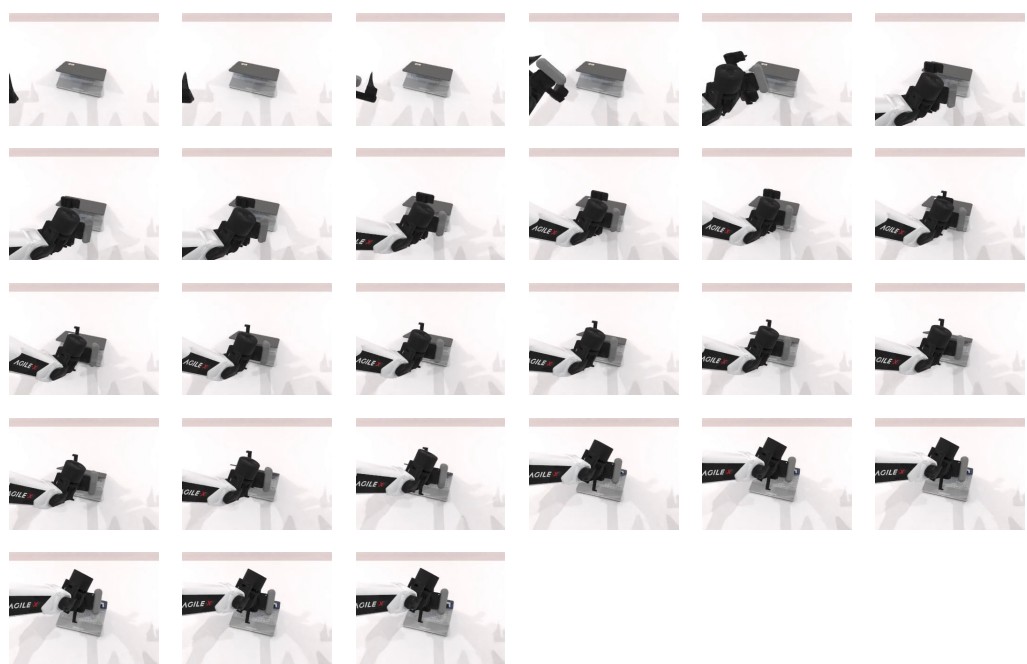

Figure 22: **Illustration of RoboTwin 2.0 Open Laptop.**

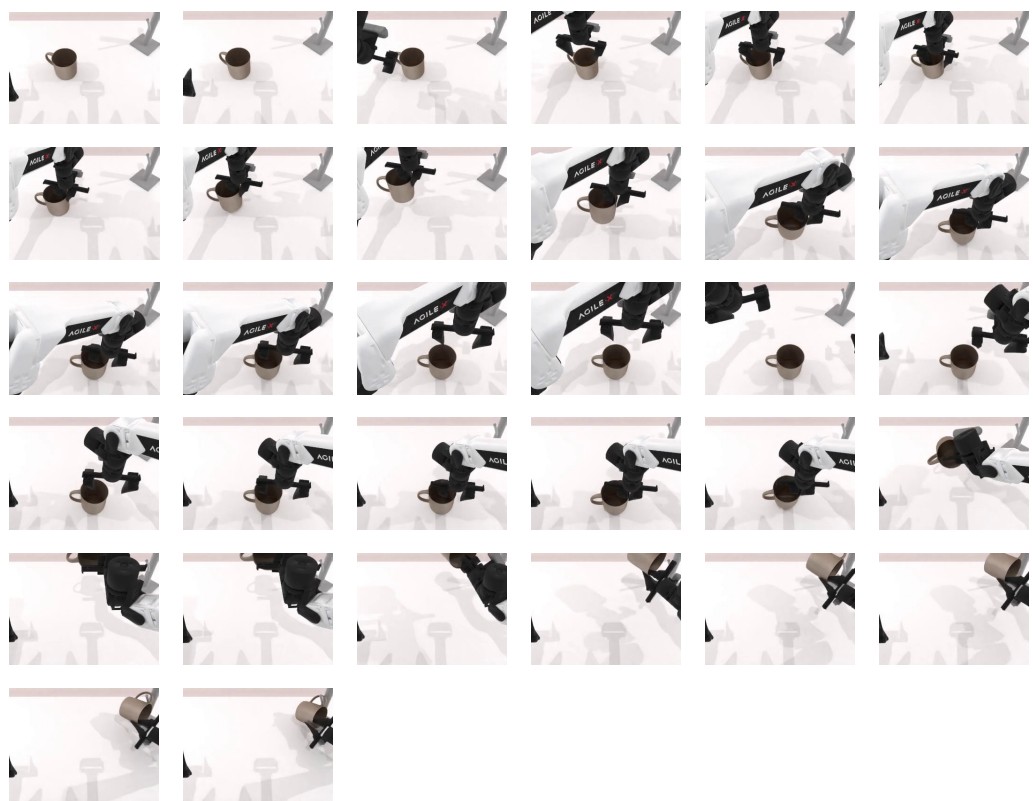

Figure 23: **Illustration of RoboTwin 2.0 Hanging Mug.**

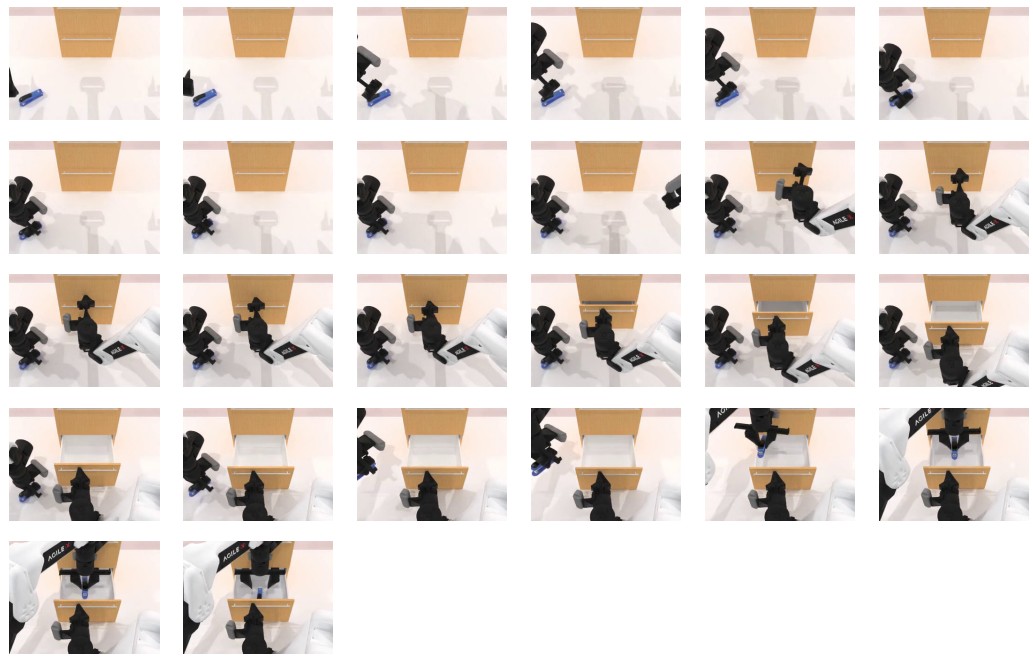

Figure 24: **Illustration of RoboTwin 2.0 Put Object Cabinet.**

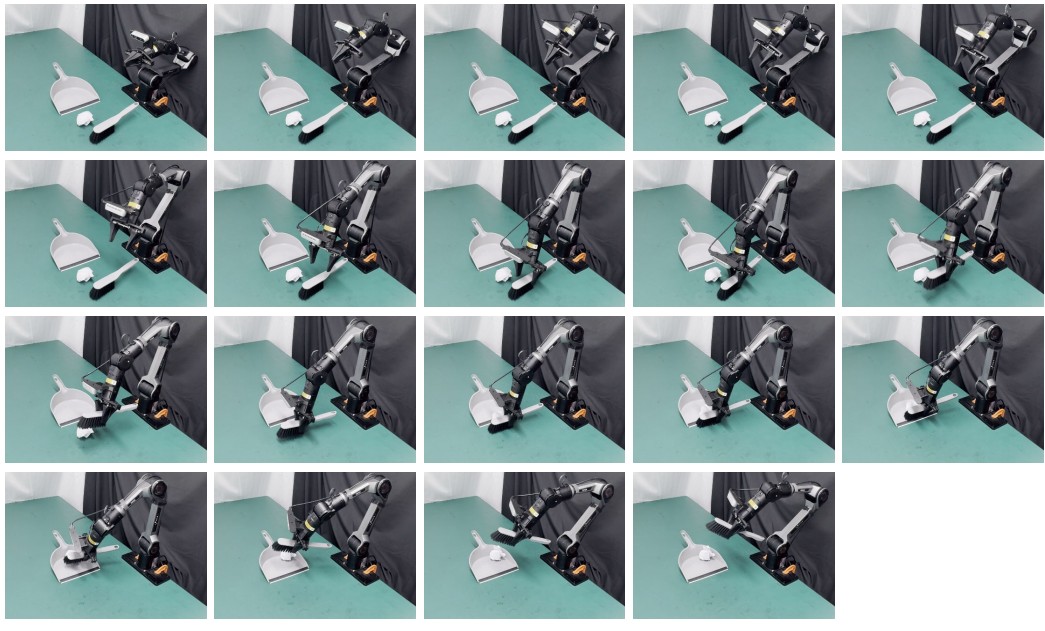

Figure 25: **Illustration of Real-world Experiment Clean Table.**

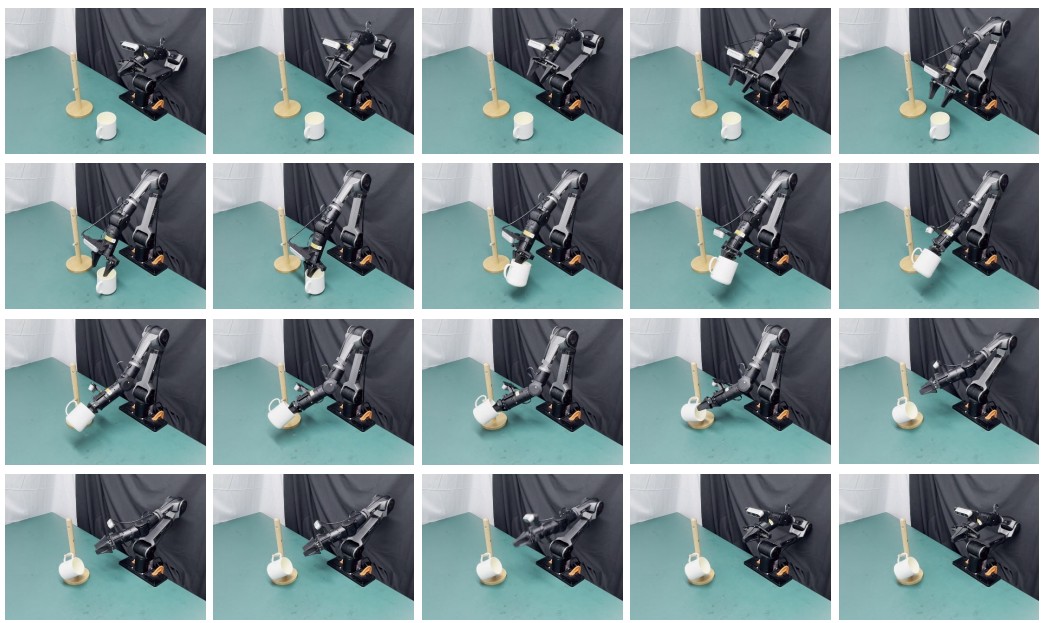

Figure 26: **Illustration of Real-world Experiment Hang Mug.**

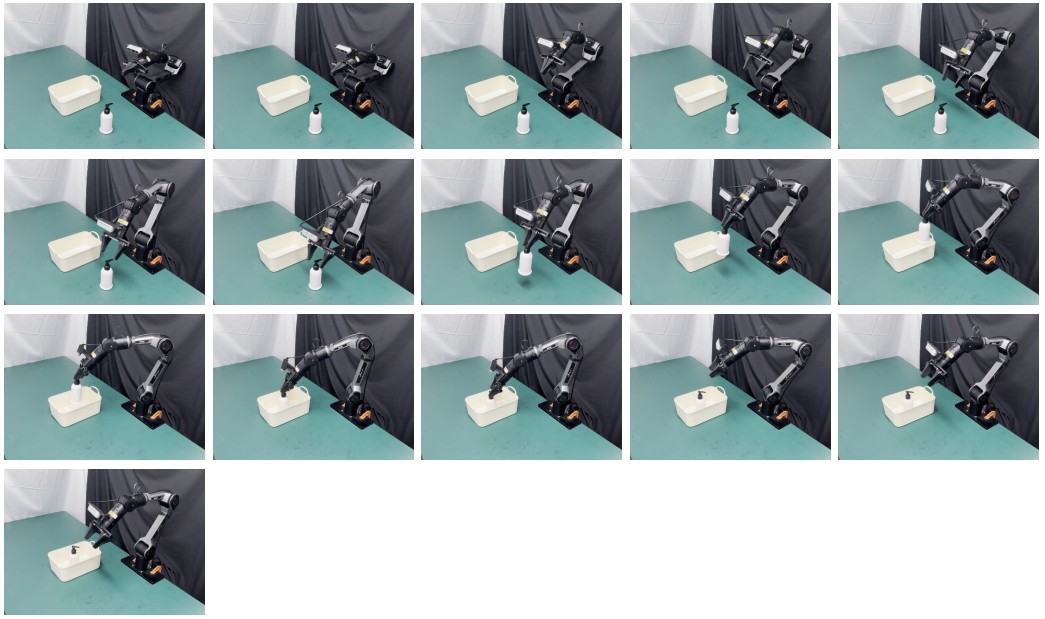

Figure 27: **Illustration of Real-world Experiment Place Bottles.**

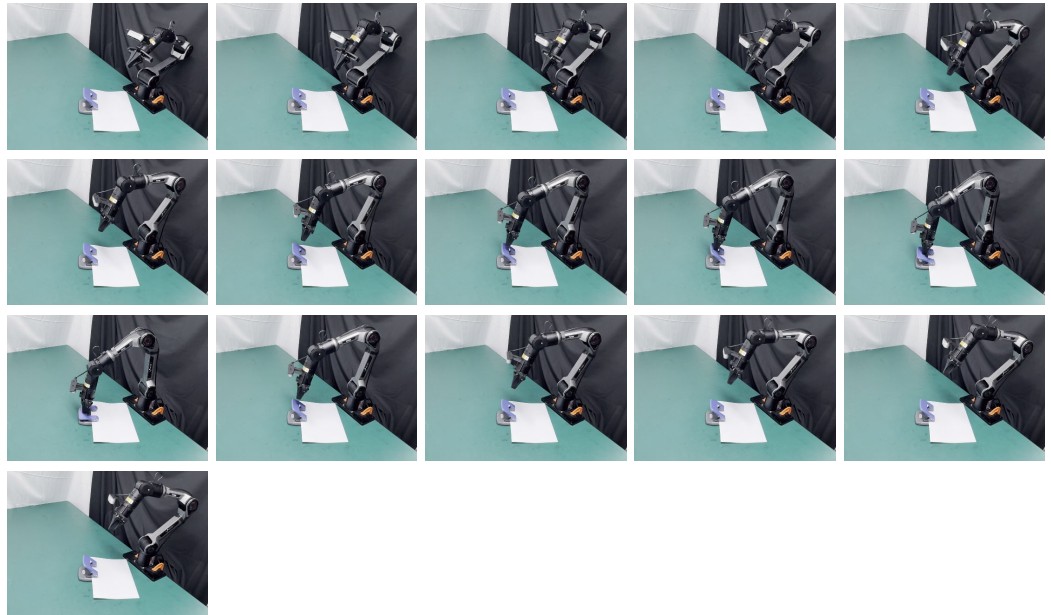

Figure 28: **Illustration of Real-world Experiment Punch Holes.**

## L    DETAILED LITERATURE REVIEW

Owing to space constraints in the main text, we expand the related work section with a detailed literature review. This appendix aims to provide a more comprehensive overview of prior work, highlighting additional studies and applications that could not be discussed in detail in the main body of the paper. All these works have made significant contributions to the robotics community.

### L.1    COMPOSITIONAL GENERATIVE MODELING

Compositional generative modeling has recently emerged as a compelling alternative to monolithic large-scale models, emphasizing the idea that complex data distributions can be captured more effectively by composing simpler factors. Instead of relying on a single, over-parameterized model, researchers argue that distributions can be factorized and modeled in a modular fashion, thereby reducing data requirements and improving interpretability (Koller & Friedman, 2009; Murphy, 2022). This line of work draws inspiration from probabilistic graphical models and energy-based formulations, but has been extended to modern deep generative architectures.

A key motivation for compositional modeling is data efficiency. By factorizing distributions into manageable components, one can achieve accurate modeling even under limited training data. For instance, Janner et al. (2022) and Ajay et al. (2022) showed that trajectory generation benefits from decomposing the sequence into components, leading to faster training and improved generalization. In natural language processing, Du et al. (2023b) demonstrated that reasoning can be improved by combining multiple large language models, effectively composing factors across models in a "multi-agent" framework. Similarly, Liu et al. (Liu et al., 2021) introduced composable diffusion for text-to-image generation, where local sentence-level factors combine to synthesize complex global scenes.

Beyond efficiency, compositionality provides a natural mechanism for generalization to novel tasks and distributions. In decision-making and planning, Ajay et al. (2023) proposed hierarchical foundation models that integrate language, video, and action policies, enabling flexible recombination for zero-shot planning (Wang et al., 2025c). In robotic manipulation, Yang et al. (2023b) and Mishra et al. (2023) showed that object rearrangement tasks can be solved by composing local constraint factors, while Wang et al. (2024c) extended this idea to heterogeneous policy composition (Wang et al., 2024b; Wang, 2025). For visual domains, Du et al. (2023a) and Zhang et al. (2023) developed methods to assemble image collages via factorized regional conditionals, while Yang et al. (2023a) demonstrated that video style transfer can be achieved by composing a pretrained prior with a lightweight style model.

Another strand of work investigates how compositional structure can be discovered automatically. Du et al. (2021) and Su et al. (2024) showed that autoencoders trained with product-of-experts likelihoods naturally uncover object-level factors, which can later be recombined to generate hybrid scenes. In dynamical systems, Comas et al. (2023) inferred relational potentials between particles, enabling recombination of discovered interaction rules. Similarly, Liu et al. (2023) found that compositional components learned on ImageNet correspond to semantic classes, making it possible to synthesize images of unseen multi-class combinations.

At the methodological level, energy-based models (EBMs) provide a natural framework for composition, since energies are additive by construction (Hinton, 2002; Du & Mordatch, 2019; Grathwohl et al., 2021). This perspective has been adapted to diffusion models, where each time step defines an implicit EBM and compositions are realized by combining energies across models (Song & Ermon, 2019; Ho et al., 2020; Du et al., 2023a). Extensions to discrete domains employ Metropolis–Hastings with learned proposals (Li et al., 2022; Verkuil et al., 2022), while Garipov et al. (2023) demonstrated how constraint energies can "sculpt" generative trajectories at inference time.

Despite these advances, challenges remain. Current approaches often assume a fixed structure of composition, limiting adaptability. Efforts by Wiedemer et al. (2023), Lachapelle et al. (2023), Misino et al. (2022), and Sehgal et al. (2023) highlight the need for robust theoretical frameworks that explain compositional generalization and offer methods to automatically infer appropriate factorization structures. Addressing these open problems will be crucial for compositional models to scale and integrate seamlessly into real-world generative systems.

## L.2 Diffusion Models in Robot Learning

Diffusion models have become a central paradigm for robot learning, offering a probabilistic framework for efficient trajectory generation and planning. Building on the recent surveys (Barreiros et al., 2025; Shao et al., 2025; Zhong et al., 2025; Xiang et al., 2025; Firoozi et al., 2025; Song et al., 2025; Sapkota et al., 2025; Wong et al., 2025; Wolf et al., 2025; An et al., 2025; Zhang et al., 2025a; Adilkhanov et al., 2025; Lin et al., 2024b; Ma et al., 2024b; Zheng et al., 2025; Liu et al., 2025b; Urain et al., 2024), we group diffusion-based robot policies into two categories (Song et al., 2025): (i) *small-size diffusion-based policies*, which integrate CNN or Transformer backbones with diffusion heads and are trained on task-specific datasets for efficient visuomotor control, and (ii) *large-scale diffusion policies*, which couple diffusion modules with pre-trained foundation models or large robot datasets to achieve broader semantic grounding and cross-embodiment generalization. Together, these developments demonstrate how diffusion can serve both as a lightweight control primitive in specialized tasks and as a scalable component in foundation-style robot policies, bridging the gap between low-level stochastic control and high-level semantic reasoning.

**Small-size CNN/Transformer-based diffusion policies.** A growing body of visuomotor research couples compact CNN or Transformer encoders with diffusion heads, showing that stochastic denoising can serve as an effective control primitive across diverse manipulation settings. Within this line, several works directly map observations to actions using diffusion: Diffusion Policy (Chi et al., 2023) established the basic recipe for action diffusion with both CNN and Transformer backbones, and DP3 (Ze et al., 2024b) extends the paradigm to point-cloud inputs to strengthen 3D spatial generalization. iDP3 (Ze et al., 2024a) extends DP3 for humanoid robots to learn from noisy human data. Mamba Policy (Cao et al., 2025b) improves DP3 by introducing a linear-complexity architecture Mamba (Gu & Dao, 2024; Dao & Gu, 2024). H$^3$DP (Lu et al., 2025) explicitly incorporates hierarchical structures to strengthen the integration between visual features and action generation. MCDP (Cao et al., 2025a) integrates DP and DP3 via compositional diffusion to achieve enhanced performances. Temporal architectures tailored for planning appear in Motion Planning Diffusion (Carvalho et al., 2023), while design ablations highlight effective Diffusion-Transformer components (DiT-Block Policy (Dasari et al., 2025)) and introduce attention-based conditioning for guided control (MTDP (Wang et al., 2025a)). Dexterous grasp synthesis is treated by DexDiffuser (Weng et al., 2024), which progressively denoises grasp configurations for multi-fingered hands. Moving beyond pooled embeddings, 3D Diffuser Actor (Ke et al., 2025) conditions on tokenized 3D scene representations, and R&D (Vosylius et al., 2024) presents a unified image–action formulation using ViT encoders. For long-horizon or structured problems, ALOHA Unleashed (Zhao et al., 2025) trains a Transformer with a diffusion loss for bi-manual skills; Chained-Diffuser (Xian & Gkanatsios, 2023) first predicts end-effector keyposes and then connects them with feasible trajectories; and HDP (Ma et al., 2024a) injects kinematics-aware priors to improve physical realism. Category-level robustness is pursued by S$^2$-Diffusion (Yang et al., 2025) via visual foundation priors for spatial semantics and by C3DM (Saxena et al., 2023) through constrained-context conditioning with a fixation step to resist distractors. Diffusion has also been adopted as a behavior prior for planning: Diffuser (Janner et al., 2022) frames sampling-based planning as probabilistic behavior synthesis, while DTP (Fan et al., 2025) adds 2D trajectory guidance in a two-stage pipeline. Language- and object-centric formulations include StructDiffusion (Liu et al., 2022), which fuses object-centric Transformers with diffusion under language goals, and PlayFusion (Chen et al., 2023), which uses discrete bottlenecks to acquire language-annotated skills. From the data side, ROSIE (Yu et al., 2023) exploits state-of-the-art text-to-image diffusion for aggressive augmentation, and Ha et al. (2023) broadens language conditioning across tasks. Finally, capacity can be increased without full monolithic scaling through expert routing: GSC (Mishra et al., 2023) probabilistically chains learned skills with classifier-based guidance to satisfy constraints, and DBC (Chen et al., 2024) stabilizes learning by casting behavioral cloning of expert state–action pairs as diffusion-based modeling.

Beyond the core diffusion-based frameworks, a number of extensions have been proposed to improve efficiency, generalization, and adaptability in visuomotor policy learning. Streaming Diffusion Policy (Høeg et al., 2024) accelerates policy synthesis by producing partially denoised trajectories where each action may retain a different noise level, while Bidirectional Decoding (BID) (Liu et al., 2024b) enables test-time inference through a combination of action chunking and closed-loop adaptation. Crossway Diffusion (Li et al., 2024b) introduces a specialized state decoder together with

an auxiliary self-supervised learning objective to reinforce policy robustness, and Equivariant Diffusion Policy (Wang et al., 2024a) exploits domain symmetries to achieve higher sample efficiency and better generalization in the denoising process. Complementary empirical work by Lin et al. (2024a) investigates data scaling effects in imitation learning at scale. Other approaches extend the representational capacity of policies, such as Imagination Policy (Huang et al., 2025a), which generates point cloud predictions of target states before converting them into executable actions, and Consistency Policy (Prasad et al., 2024), which distills faster visuomotor policies through a consistency regularization process. For fine-tuning, DPPO (Ren et al., 2024) provides a unified framework that integrates policy gradient techniques with diffusion policies in continuous control domains. Extensions to tactile-rich scenarios include the Reactive Diffusion Policy (Xue et al., 2025), which combines slow-fast visual–tactile imitation learning for contact-rich manipulation.

Several recent efforts also integrate reasoning and sequence modeling into the diffusion paradigm. The Unified Video Action Model (Li et al., 2025) jointly optimizes video prediction and action inference for accurate and efficient trajectory generation, while Chain-of-Action (CoA) (Zhang et al., 2025b) explicitly reasons backward from task goals, producing coherent trajectories through an action-level chain-of-thought mechanism. Beyond diffusion, flow matching has emerged as a strong alternative: ManiFlow (Yan et al., 2025) combines flow matching with consistency training to synthesize dexterous actions in just one or two steps; Flow Matching Policy Gradients (McAllister et al., 2025) embed flow matching directly into policy gradient algorithms for reinforcement learning; and VITA evolves latent visual states into latent actions under a flow matching framework; Steering Your Diffusion Policy (Wagenmaker et al., 2025) adapts behavior-cloning policies by performing reinforcement learning over the latent noise space, offering a flexible way to guide policy behavior without retraining from scratch. In addition to these extensions, recent works also explore safety and dynamics-aware guidance. DynaGuide (Du & Song, 2025) introduces a steering mechanism for diffusion policies by incorporating feedback from an external dynamics model directly into the denoising process, enabling more physically consistent action generation. Latent Policy Barrier (LPB) (Sun & Song, 2025), inspired by control barrier functions, formulates expert latent embeddings as implicit safety boundaries that distinguish in-distribution states from out-of-distribution ones, thereby enhancing robustness in visuomotor policy learning.

Together, these results indicate that lightweight diffusion policies augmented by stronger scene encoders, subgoal scaffolding, data augmentation, or MoE-style (Jiang et al., 2024) routing are competitive and data-efficient when embodiment and task distributions are relatively constrained.

**Large-size LLM–based diffusion policies.** At larger scales, diffusion modules are integrated with pre-trained vision–language-model (VLM) or LLM backbones or trained atop broad cross-embodiment corpora, marrying semantic understanding with probabilistic action generation. Methods leveraging general data pre-training use foundation models to inject world knowledge and linguistic structure: MDT (Reuss et al., 2024) builds on CLIP (Radford et al., 2021) and Voltron (Karamcheti et al., 2023) for long-horizon manipulation with sparse language, enriching instructions via GPT-4 (Achiam et al., 2023); Ha et al. (2023) employs LLMs for high-level plan synthesis and success inference while delegating low-level control to diffusion policies; ROSIE (Yu et al., 2023) uses LLM-authored prompts to drive text-to-image diffusion for targeted data augmentation; and TinyVLA (Wen et al., 2025a) freezes a multimodal backbone and applies parameter-efficient tuning ($\sim$5% trainable) to produce actions efficiently. Compositional planning stacks further tighten the loop between language and diffusion: HiP (Ajay et al., 2023) composes expert models LLMs for task planning, video diffusion for trajectory proposals, and an inverse model for action mapping, while Plan Diffuser (Sharan et al., 2024) autoregressively emits textual subgoals with an LLM and translates them into visual subgoals via diffusion for downstream control. In parallel, robot data pre-training focuses on large, heterogeneous robot datasets to strengthen embodiment transfer. Octo (Team et al., 2024) aggregates 25 datasets from Open X-Embodiment (O'Neill et al., 2024) and trains a Transformer with a diffusion head to map observation/task tokens to action tokens across embodiments; Diffusion-VLA (Wen et al., 2025b) pre-trains on Open X-Embodiment and DROID (Khazatsky et al., 2024) and adapts to tasks via LoRA (Hu et al., 2022); ChatVLA (Zhou et al., 2025) co-trains on robot and reasoning data with staged alignment and MoE routing to reduce task interference; RDT-1B (Liu et al., 2024a) specializes in fine-grained (including bimanual) skills by standardizing a unified action space over heterogeneous robots; and LAPA (Ye et al., 2024) systematically studies cross-embodiment pre-training using BridgeV2 (Walke et al., 2023) and Open X-Embodiment (O'Neill et al., 2024). $\pi_0$ (Black et al., 2024) couples a pre-trained

vision–language backbone with a flow-matching action expert for precise, smooth manipulation; Based on $\pi_0$, $\pi_{0.5}$ (Intelligence et al., 2025) uses co-training on heterogeneous tasks to enable broad generalization; Pertsch et al. (2025) and Driess et al. (2025) focused on accelerating the VLAs based on empirical studies; Enerverse (Huang et al., 2025b) and Video Prediction Policy (Hu et al., 2024) use diffusion models for learning visual representations to improve scene understanding and subsequently enhance policy performance; HybridVLA (Liu et al., 2025a) unifies both the continuous nature of diffusion-based actions and the contextual reasoning of autoregression within a single LLM; GR00T N1 (Bjorck et al., 2025) is a dual-system (Figure, 2024; Cui et al., 2025) VLA for generalist humanoid robots, achieving state-of-the-art performances across multiple robot embodiments; Galaxea G0 (Jiang et al., 2025) also adpots the dual-sytem, coupling a VLM for multimodal planning and a VLA for low-level robot control; Agibot GO-1 (Bu et al., 2025) is a generalist policy that leverages latent action representations to maximize data utilization; Gemini Robotics family (Team et al., 2025) achieves generalized abilities in diverse tasks, including robot control, object detection, pointing, trajectory, and grasp prediction. In addition to using the diffusion policy as the action head, there are many excellent works that employ flow matching (Lipman et al., 2023; Liu, 2022) for action prediction: GraspVLA integrates autoregressive perception tasks and flow matching-based action generation into a unified Chain-of-Thought process; GR-3 (Cheang et al., 2025) excels in understanding complex instructions with abstract concepts, generalizes effectively to novel objects and environments; WALL-OSS (Zhai et al., 2025) presents a coupled architecture, unifying instruction reasoning, subgoal decomposition, and fine-grained action synthesis.

Overall, these large-scale policies suggest a convergent recipe in which language models structure objectives and subgoals, diffusion processes synthesize trajectories and actions, and pre-training (on general or robot-centric corpora) supplies the semantic and embodiment priors required for robust generalization.

### L.3 NON-DIFFUSION-BASED MODELS IN ROBOT LEARNING

While diffusion-based approaches have recently attracted significant attention, a wide range of non-diffusion architectures continue to play a pivotal role in robot learning. These models typically leverage sequence modeling, spatial reasoning, and large-scale VLA systems to build versatile manipulation and navigation capabilities. Unlike diffusion policies, which rely on iterative denoising, non-diffusion frameworks often emphasize direct policy learning through MLP or transformers (Vaswani et al., 2017), hierarchical control, or data scaling strategies. In what follows, we summarize representative advances in manipulation and navigation, highlighting how non-diffusion models complement and extend the landscape of robot learning.

**Manipulation.** A major thread in non-diffusion visuomotor learning frames control as sequence modeling. ACT (Zhao et al., 2023) introduces a conditional encoder–decoder Transformer that predicts action sequences rather than single steps, attenuating compounding error over long horizons. Building on this idea, MT-ACT (Bharadhwaj et al., 2024) augments training with task semantics to learn a universal multi-task manipulator, while CogACT (Li et al., 2024a) couples a VLA backbone so that language-guided cognition and low-level motor control are optimized in concert. Chunking Causal Transformer (Zhang et al., 2025c) retains the ACT-style autoregressive policy but segments trajectories into chunks, improving stability and sample efficiency for long sequences. Beyond pure sequence decoding, several works enrich spatial grounding and 3D control: Act3D (Gervet & Xiao, 2023) is a language-conditioned Transformer for 6-DoF manipulation that outputs continuous-resolution 3D action maps via adaptive 3D computation; ICRT (Fu et al., 2024) performs genuine in-context learning on a physical robot, leveraging a handful of contextual trajectories to execute unseen tasks without additional training; and Spatial Policy (Liu et al., 2025c) explicitly models scene geometry so that visual predictions align with executable end-effector motions. A complementary line scales VLA systems with data and hierarchy: RT-1 (Brohan et al., 2023) demonstrates that Transformers trained on large, diverse robot datasets yield strong generalists; RT-2 (Zitkovich et al., 2023) transfers web-derived vision–language knowledge into control; RT-X (O'Neill et al., 2024) shows that pretraining on large-scale OXE data can set new performance bars, underscoring the value of data scale; and RT-H (Belkhale et al., 2024) inserts a language-motion layer that bridges high-level instructions and low-level actions through an explicit hierarchy. Practical systemization and modality breadth are advanced by Beyond Sight (Jones et al., 2025) (Octo-style finetuning to adapt generalist visuomotor policies to heterogeneous sensors), OpenVLA (Kim et al., 2025) (fully

released training/testing recipes), and RoboVLM (Li et al., 2024c) (a design study distilling the most consequential choices in VLA pipelines). Finally, emerging embodied models lift perception and coordination to 3D and dexterous settings: 3D-VLA (Zhen et al., 2024) links 3D perception, reasoning, and action via a generative world model; Bi-VLA (Gbagbe et al., 2024) targets coordinated bimanual manipulation; LEO (Huang et al., 2024) acts as a multimodal generalist capable of perceiving, grounding, reasoning, planning, and acting in 3D environments; SpatialBot (Cai et al., 2025) strengthens spatial understanding by fusing RGB with depth; Lift3D (Jia et al., 2024) elevates 2D foundation features into robust 3D manipulation representations; and RoboDual (Bu et al., 2024) unifies generalist breadth with specialist precision in a synergistic dual-policy framework.

Further advances focus on constraint-driven representations for manipulation. Relational Keypoint Constraints (ReKep) (Huang et al., 2025c) define visually grounded constraints as Python functions that map sets of 3D keypoints in the scene to numerical costs, providing a flexible interface for encoding task-specific relations. VosPoser (Huang et al., 2023) leverages large language models to extract affordances and constraints from natural language, composing 3D value maps in the observation space that guide robotic interactions in a structured manner.

Together, these works illustrate that non-diffusion architectures, particularly sequence models and VLA systems, achieve strong manipulation generalization through long-horizon decoding, explicit spatial grounding, data scaling, and modular hierarchy.

**Navigation.** For locomotion and navigation, non-diffusion approaches similarly exploit hierarchy, distillation, and language grounding. Cheng et al. (2024b) develop extreme legged parkour by first training a teacher with reinforcement learning and then distilling its competence into a student policy that runs purely on onboard depth, enabling agile behaviors in the wild. Mobility VLA (Chiang et al., 2024) adopts a hierarchical design: long-context VLMs provide scene understanding and commonsense reasoning at the high level, while a robust low-level navigator follows a topological graph to execute the plan. NaVid (Zhang et al., 2024b) turns streaming RGB video and a natural-language instruction into a sequence of textual action directives that a robot can carry out, emphasizing language-as-action for purely visual inputs. NaVILA (Cheng et al., 2024a) extends this idea to legged visual–language-navigation (VLN) with two levels of control: a finetuned VLM outputs mid-level language actions (*e.g.*,"turn right 30"), and a learned visual locomotion controller faithfully executes those commands. These systems highlight a recurring pattern in non-diffusion navigation: decompose high-level intent into compact linguistic subgoals, pair them with robust low-level policies for accurate robot control.

## M  FUTURE WORK

This work opens several avenues for future exploration. On the methodological side, a key direction is to move beyond fixed test-time weight discretization. More adaptive weighting strategies could be developed, such as reinforcement learning or gradient-based meta-optimization, to automatically adjust convex weights across tasks and environments. Another natural extension is to scale from dual-policy to multi-policy composition. Since naïvely increasing the number of composed policies incurs a high computational cost, future work may explore feature sharing mechanisms or compact latent representations to enable efficient integration. Finally, the design of stronger composition operators remains an open challenge. Our initial results with superdiffusion highlight its potential, but more efficient variants, as well as extensions that integrate with flow-based models, could further amplify policy performance.

At a broader level, the principle of policy composition can potentially extend beyond diffusion-based policies. The same compositional framework could be applied to diverse policy classes and architectures, enabling modular integration of heterogeneous skills. Moreover, while our experiments focus on robotic VA and VLA in manipulation tasks, we anticipate a broader impact in related domains. For instance, vision-language-navigation (VLN) tasks, such as some successful state-of-the-art methods TrackVLA (Wang et al., 2025b) and LOVON (Peng et al., 2025), may also benefit from compositional strategies to enhance generalization and robustness. Exploring these directions would further validate GPC as a general paradigm for leveraging pre-trained models in complex sequential decision-making domains.

