# OpenReview forum: "Compose Your Policies! Improving Diffusion-based or Flow-based Robot Policies via Test-time Distribution-level Composition"
_ICLR.cc/2026/Conference — ICLR 2026 Poster_

### Official Review · Reviewer_d2UQ · 2025-10-31

**Soundness:** 3
**Presentation:** 3
**Contribution:** 3
**Rating:** 6
**Confidence:** 2

**Summary:**

This paper propose a policy composition methods in imporving the policy performance in the inference time without any data collection or training. This paper conducts comprehensive experimentis in several benchmarks and real world experiments with various base policies, together with detailed theoretical analysis.

**Strengths:**

1. This paper proposed a novel framework in composing different policies during inference time without collecting more data or training.

2. The proposed method can handle different policies, datasets, and modalities for a task under the same environments.

3. This paper conducts comprehensive experiments and theortical analysis accross several benchmarks and tasks with various different policies.

**Weaknesses:**

1. Combining different policies and modalities will increase the computation cost during inference time.

2. The output actions should be the same. But in general, for different kinds of policies like DP, VLA, and flow-based policy might be good at using different length of action chunkings. It might be a limitation.

3. The ideas, results, and analysis are interesting, but not quite sure this method is pratical or not... To compose them and get performance, I need to train different policies for a same task, and even collect different data with different modalities. With the same cost of time, collecting more data can also increase the preformance.

**Questions:**

No questions

---

> ### Author Response · Authors · 2025-11-18
> **Official Response**
>
> We greatly appreciate your thoughtful and constructive comments and have reflected your suggestions in the revised version of the paper! Our responses are as follows:
>
> > **Q1)** Inference computation cost of GPC
>
> We agree that combining multiple policies and modalities inevitably increases inference-time computation, since more than one model must be queried per control step. In practice, however, we find that this overhead is **modest**.
>
> We measure the per-action-chunk inference time in Robomimic as follows:
>
> | Method | Time per action chunk |
> | - | - |
> | DP    | 0.09s    |
> | FP-D     | 0.06s     |
> | GPC    |  0.13s   |
>
> Thus, composing two policies increases per-step latency by only a few tens of milliseconds in our setup, and in our real-robot experiments GPC still meets real-time control requirements. Since this overhead is purely computational, it can be further reduced with stronger hardware (e.g., better GPUs, accelerators) and optimized inference runtimes.
>
>
>
> > **Q2)** Can GPC still work in different action chunk conditions?
>
> GPC can explicitly handle heterogeneous chunk sizes by composing scores on the overlapping part and letting the longer-chunk policy govern the rest.
>
> Concretely, let $H_A \ge H_B$ be the chunk lengths of policy A and B. At a given noise level, we:
> 1. Sample a shared noise trajectory of length $H_A$.
> 2. Query policy A on the full trajectory to obtain $\mathbf{s}_A \in \mathbb{R}^{H_A \times d}$.
> 3. Query policy B on the prefix (first $H_B$ steps) to obtain $\mathbf{s}_B \in \mathbb{R}^{H_B \times d}$.
> 4. Compose the scores as: $\mathbf{s}_{\text{comp}}[1{:}H_B] = w_1 \mathbf{s}_A[1{:}H_B] + w_2 \mathbf{s}_B$,  and use $\mathbf{s}_A$ for the remaining tail.
>
> This corresponds to sampling from a composite distribution $q(\mathbf{a}) \propto p_A(\mathbf{a})^{w_1}p_B(\mathbf{a})^{w_2} ,$
> where B only shapes the shared prefix while A provides the full-horizon structure. The resulting trajectory is jointly determined by the composed prefix score and the longer-chunk tail, which is a **non-trivial** combination rather than a simple truncation. This mixed-length score composition is a new design introduced in this work. Our core theory is developed for equal-length chunks, and the heterogeneous-chunk setting goes beyond that theoretical setup.
>
> We therefore treat it as an extended case, validate it empirically with the design above on RoboMimic (Can), where the two base policies use different chunk lengths:
>
> | Method                     | Settings                                                  | RoboMimic Can |
> |----------------------------|-----------------------------------------------------------|---------------|
> | Diffusion Policy           | action chunk size = 8                                     | 0.50          |
> | Florence-Policy-D         | action chunk size = 16                                    | 0.53          |
> | GPC (DP + Florence-D)      | action chunk size = 16 | **0.68**      |
>
> These results show that GPC can **benefit from combining policies with different preferred chunk lengths** instead of being restricted by them. We will clarify this in the revised version and highlight heterogeneous chunking as an interesting and non-trivial setting for future theoretical and empirical exploration.
>
>
> > **Q3)** Practical Use of GPC compared with collecting more data
>
> GPC is designed for a different, more realistic regime:
>
> - **Reuse existing policies or light finetuning**. Modern robotics already offers many strong pretrained checkpoints, e.g., $\pi0$. In our experiments, we reuse such policies and, when needed, only apply light task-specific finetuning with on the order of ~100 demonstrations per task, rather than collecting the massive datasets typically required to significantly improve a single large VLA via pure data scaling. Given the high cost of real-robot data, “just collect much more data for one model” is often not feasible, whereas composing a few reasonably strong policies can be more viable.
>
> - **Forward-compatible with future generalist VLAs.**  Looking ahead, if robotics reaches an LLM-like stage with strong off-the-shelf VLAs, many tasks may no longer require any finetuning. In that regime, GPC becomes a **pure inference-time mechanism**, combining multiple generalist policies with no extra training or data collection cost, only modest additional inference cost.
>
> We have clarified in the revised version (App. O) that GPC is intended as a practical way to leverage existing policies under limited data/compute, and will only become more attractive as stronger generalist VLAs emerge.
>
> ### Conclusion
> Thank you again for your thoughtful feedback! We have updated the manuscript accordingly and hope that our revisions and additional analyses satisfactorily address your questions.

---

> ### Author Response · Authors · 2025-11-27
> **Gentle Reminder**
>
> Hello! Thank you for your valuable suggestions. We’ve addressed the issues raised in your review and updated the revised version accordingly. If you think any part could use additional clarification or expansion, we’d be delighted to discuss.
>
> Kind regards,
>
> The Authors

---

### Official Review · Reviewer_mnxH · 2025-11-03

**Soundness:** 3
**Presentation:** 3
**Contribution:** 2
**Rating:** 4
**Confidence:** 3

**Summary:**

This paper introduced a training-free method for boosting diffusion-based policies performance via test-time policy composition. Theoretical analysis is provided to support the claim that convex combination of different policy distribution can improve functional objective. The proposed General Policy Composition (GPC) has been validated on various simulation benchmarks and real-world robot system.

**Strengths:**

1. The idea of training-free test-time policy improvement with policy composition is novel and interesting
2. The theoretical proof provides good mathematical foundation for the proposed method (I have not thoroughly read the proof in the Appendix, may require extra verification)
3. Detailed analysis and ablations are conducted in both simulation and real-world experiments.
4. The paper is well-written in general and easy to follow.

**Weaknesses:**

1. Based on the results, it appears that the proposed method yields a noticeable performance improvement only when both base policies perform well. As noted in Section 6.3, if either policy performs poorly to begin with, the proposed method tends to struggle and may even degrade overall performance.
2. The performance of GPC is highly sensitive to weight tuning, which limits its practical generalizability.
3. GPC’s effectiveness relies on the diversity of modalities offered by the base policies; however, there is no guarantee that such diversity exists, nor is there any further analysis on how varying modalities influence the final performance.

**Questions:**

1. GPC has an implicit assumption that two base policies should have different behavior distributions, how to make sure this is the case in practice?
2. How much extra inference cost is required for GPC?
3. In case of VA+VLA, how to guarantee the VA modality correctly follow the language instruction of VLA? If VA's modality is opposite to the language instruction, will GPC in fact harm the final performance?

---

> ### Author Response · Authors · 2025-11-18
> **Official Response**
>
> Thank you for your thoughtful feedback! We have revised the paper accordingly. Here are our responses:
>
> > **Q1)** Better results when composing strong policies
>
> We agree with the finding: GPC tends to yield the largest gains when both policies are strong, and can struggle when one is very weak. We believe this behavior is **natural** and, in fact, expected for any method based on combinations of model abiliy rather than a specific flaw of GPC.
>
> If one base policy is very poor and is given a large weight, the convex combination can be worse than the best individual score. This is **unavoidable** for most ensemble-like methods (e.g., simple averaging, mixture-of-experts and classifier-free guidance).
>
> Practically, we view this not as a hidden failure mode but as **important usage guidance**. Sec. 6.3 studies this effect and shows that performance is maximized when the **better-performing base policy receives a larger weight**, especially in asymmetric strong+weak settings. This leads to a recommendation: apply GPC to reasonably competent policies and bias the weights toward the stronger one. As the ecosystem of strong policies grows, we expect the benefits of GPC to become even more prominent.
>
> > **Q2)** Determining the weighting factor of two policies
>
> We agree that the weights in GPC are an important factor and treat them as **hyperparameters**, analogous to the guidance scale in classifier-free guidance for diffusion models: performance is sensitive to this scale, yet the community has converged on useful default ranges from empirical studies.
>
> In the same spirit, our weight sweeps are reported to **elucidate** this sensitivity, provide concrete guidance, and yield simple practical rules (e.g., biasing weights toward the stronger policy) that already make GPC usable in practice and suggest reasonable default ranges for future users.
>
> > **Q3)** GPC’s effectiveness with the diversity of modalities
>
> We would like to clarify that **GPC does not rely on modality diversity as a requirement**. GPC only assumes that each base policy defines a valid score, regardless of whether they use the same or different modalities. Our result highlights a cross-modal case, importantly, we also observe consistent gains in **same-modality** settings where policies use RGB only but differ in arch, e.g., Pi-0 + Flow Policy. These results show that GPC can be effective even with no additional sensor modality, **by leveraging architectural and representational diversity** instead.
>
> > **Q4)** Behavioral diversity with GPC
>
> Intuitively, GPC benefits most when the base policies exhibit *some* behavioral diversity, but this is not a brittle or hard assumption in GPC.
>
> If two policies were truly identical and produced the same score field everywhere, then composing them would simply recover that same score, GPC would neither help nor hurt. In contrast, when policies differ in their strengths, the convex composition can yield a better score. This is analogous to standard ensemble methods: diversity is a source of improvement, not a strict requirement for correctness.
>
> In practice, **behavioral diversity arises naturally** without requiring any artificial construction. In our work, we deliberately combine policies that differ either in:
> - **Modality** (e.g., DP(img) + DP(pcd) )
> - **Architecture** (e.g., $\pi0$ + Flow Policy)
>
> In highly non-convex policy learning, such differences in input modality and backbone almost inevitably lead to distinct behavior distributions, which is what we observe empirically.
>
> > **Q5)** Inference cost for GPC
>
> We provide the inference time comparison below. GPC only adds a small overhead compared to running a single policy, where it can be further reduced with stronger hardware settings.
>
> | Method | Time per action chunk |
> | - | - |
> | DP    | 0.09s    |
> | FP-D     | 0.06s     |
> | GPC    |  0.13s   |
>
> > **Q6)** Alignment between VA's modality and text
>
> In our setting, the VA policy is **not** trained on a task distribution that is “opposite” to the language instructions. Both VA and VLA are trained on demonstrations from the *same family of tasks*:
> - VLA: $\pi(a \mid o, \text{text})$ conditioned on language,
> - VA: $\pi(a \mid o)$ learned on the **same successful trajectories**, but without text.
>
> Thus, the VA serves as a language-agnostic *skill prior* over reasonable behaviors, rather than a policy that contradicts the instruction. In our experiments, this prior is broadly aligned with the tasks described by the language, even though it does not explicitly see the text at inference time.
>
> We agree that if a VA policy were truly misaligned with the instructed task and still given a large weight, it could harm performance. This is a general property of any ensemble that includes a strongly misaligned expert, not a GPC-specific mode.
>
> ### Conclusion
> We are grateful for your suggestions which helped us improve the paper! We would be happy to discuss any remaining concerns!

---

> ### Author Response · Authors · 2025-11-27
> **Gentle Reminder**
>
> Hi! Many thanks again for the constructive feedback. We’ve resolved the concerns you mentioned and reflected all changes in the revised version. If anything else comes to mind or you’d like to explore certain points further, we’re more than happy to chat.
>
> Best,
>
> The Authors

---

### Official Review · Reviewer_vsK6 · 2025-11-03

**Soundness:** 4
**Presentation:** 3
**Contribution:** 3
**Rating:** 6
**Confidence:** 3

**Summary:**

This work presents General Policy Composition (GPC), a training-free framework designed to enhance the performance of robotic control policies, particularly those based on diffusion or flow-matching models, by combining existing pre-trained policies. GPC addresses the high cost and data requirements typically associated with scaling up or fine-tuning control policies. The core mechanism involves composing the distributional scores of two policies during inference, via a convex combination, utilizing a test-time search to identify optimal weighting coefficients. There is a theoretical foundation to show that a convex combination of scores yields a provably superior one-step functional objective compared to individual scores, and this improvement propagates throughout the entire generation trajectory.

Empirical validation across simulation benchmarks (Robomimic, PushT, RoboTwin) and real-world tasks shows that GPC improves performance, achieving average success rate increases of up to +7.55% in sim and +10% in real-world tasks compared to single-policy baselines. The author's analysis suggests that GPC generates more coherent and concentrated distributions throughout execution, contributing to stability. Although effective, the framework is currently limited by reliance on a fixed discretization for test-time weight search and potential computational increases when scaling beyond dual-policy composition.

**Strengths:**

1. Simple, effective, and theoretically-motivated idea that is reminiscent of other techniques in ML that use multiple models to get the best outcome (e.g. MoE, boosting).
1. Idea of utilizing test-time search is sound; it would help the authors to add references talking about the per-compute-unit performance benefits of using test-time compute vs. train-time compute. (e.g. arxiv.org/pdf/2408.03314).
1. The proposed GPC framework enables composition across both VA and VLA models.

**Weaknesses:**

1. Potentially very large inference time due to weight search. Already, robotics policies struggle with inference that is not fast enough to keep up with physics.
1. No major comparison of how much test-time compute is used against the compute used for the base policies that were composed.
1. No discussion about a potential increased test-time inference latency.

**Questions:**

1. How would you address giant compute overheads for running a VA/VLA model while trying to meet the real-time requirements of running policies on a real robot?
1. Fig 1: Explain the desired task here in the caption. Preferably give the task prompt or description that is input to the policies.
1. Line 361 (100-200 rollouts for evals): I’m trying to understand how often this test-time compute is calculated during rollout. What’s the obs horizon over which these trajs are created? Is it one DP inference call per episode?
1. Sec 6.2 and table 1: What is the reason for some combinations to exhibit a higher performance increase? Please give your analyses and hypotheses, if any. How much of this improvement would you attribute to causes such as choice of task, choice of underlying models, choice of modalitiies for each model, randomness during evals, and any other cause you would like to highlight.

---

> ### Author Response · Authors · 2025-11-18
> **Official Response**
>
> We sincerely thank you for the thoughtful and constructive feedback. We have incorporated these suggestions and the recommended citation into the revised version of the paper. Our responses are as follows.
>
> > **Q1)** Weight search and real-world deployment
>
> We would like to clarify that we perform weight search only once for each task, by evaluating the success rate at several fixed convex combinations of the parent policies. Once this search concludes, we fix the selected weights and use them for all future rollouts of the composed policy. Thus, for online control, GPC only increases inference time for querying 2 base policies per action.
>
> In our real-world setting, using RTX 6000 Ada with PIPER, this additional cost does **not** prevent the policy from keeping up with the physics, and the tasks can be smoothly executed.
>
> More broadly, we view inference efficiency as an engineering question that is orthogonal to our work: as hardware and deployment stacks improve (e.g., better GPUs), both VLAs and GPC will simultaneously benefit.
>
> > **Q2)** Inference speed comparision & real-time requirements
>
> We report the inference speed in RoboMimic below, showing that the additional per-action compute introduced by GPC is **modest** in practice.
>
> | Method | Time per action chunk |
> | - | - |
> | DP    | 0.09s    |
> | FP-D     | 0.06s     |
> | GPC    |  0.13s   |
>
> We agree that addressing compute overheads with large VLAs is an important challenge for the entire robotics community. In fact, many recent systems already operate at relatively low infer frequencies (e.g., OpenVLA & RDT runs at 6Hz, DP runs at 10 Hz), yet still successfully complete a wide range of real-world tasks. Our deployment follows a similar regime and GPC is also able to meet real-time requirements to complete the tasks.
>
> There are multiple complementary directions to further reduce latency, e.g., using stronger hardware, or techniques like streaming outputs [[1]](https://arxiv.org/pdf/2406.04806), or distillation [[2]](https://rdt-robotics.github.io/rdt2/). Any advance that makes inference faster will also directly benefit GPC.
>
> > **Q3)** Task prompt in experiments
>
> Here we describe the text prompts used as input to the VLA policies.
>
> For the RoboTwin “place burger”, the task description: “Use dual arm to pick the hamburg and french fries and put them onto the tray,” with schema: “{A} denotes the hamburg, {B} denotes the tray, {C} denotes the french fries.”
>
> During training, the VLA receives diverse paraphrased prompts, e.g., “Use both arms to move {A} and {C} to {B}”; “Lift {A} and {C}, placing them neatly on {B}.”
> At test time, we use unseen prompts, e.g., “Pick {A} and {C}, then place them on {B}.”
>
> > **Q4)** Detailed test-time compute and obs horizon
>
> In our paper, a rollout refers to: from env reset until the task terminates in success/failure. An evaluation then consists of running $N_{\text{rollout}}$ and computing the avg success rate. “100–200 rollouts for evals” means: RoboTwin sets $N_{\text{rollout}} = 100$ rollouts per eval; RoboMimic / PushT set $N_{\text{rollout}} = 200$.
>
> Within each rollout, the policy is queried constantly. At inference time step ${t}$, the policy receives the current & history obs {$o_{t-k}, ..., o_{t}$} (the obs horizon depends, RDT set it as 2, DP sets it as 3) and outputs an action chunk {$a_{t}, ..., a_{t+q}$}, which the robot executes; the resulting new state is then fed back to the policy at the next step.
>
> > **Q5)** Reasons for good performances
>
> Intuitively, GPC can be viewed as forming a **product** of probability density functions: A higher weight on one policy means its density contributes more strongly to the final target distribution, concentrating probability mass on trajectories that both experts consider likely.
>
> (1) Theoretically:
>
> Prop. 1 shows that at each diffusion step, there exists a convex weight such that the composed score is closer to the ideal score $s^*$. Prop. 2 extends this along the full trajectory, implying that once per-step score quality improves, the resulting trajectory distribution is better.
>
> (2) Empirically:
> - **Complementary modalities.**  Combinations that fuse different modalities tend to yield larger gains, since the composed score exploits richer and more complementary information of the scene.
> - **Diverse architectures.** Different backbones encode different inductive biases and error patterns. GPC aggregates their strengths while averaging out idiosyncratic mistakes.
> - **Task- and model-specific strengths.** GPC works best when the stronger policy receives a larger weight, explaining why certain task–model pairs and weight settings exhibit bigger improvements.
> - **Weight selection**. Our weight-sweep studies illustrate how the choice of weights shapes the composed distribution.
>
> ### Conclusion
>
> We sincerely appreciate your insightful comments and have incorporated the suggested changes into the revised version. We hope our responses address your concerns!

---

> ### Author Response · Authors · 2025-11-27
> **Gentle Reminder**
>
> Hello! Thank you once more for your helpful comments. We’ve worked through the issues you pointed out and added the updates to the revised manuscript. Should you have any additional thoughts or feel that further details would be useful, we’d be glad to discuss them.
>
> Warm regards,
>
> The Authors

---

### Official Review · Reviewer_BM4P · 2025-11-04

**Soundness:** 3
**Presentation:** 3
**Contribution:** 2
**Rating:** 4
**Confidence:** 4

**Summary:**

The paper proposes General Policy Composition (GPC), a training-free method that improves robotic policies by composing distributional scores from multiple pre-trained diffusion/flow policies at test time. GPC performs a test-time weight search over convex coefficients of two score estimators, then samples trajectories using the composed score. Experiments on a simulated manipulation benchmark and on multiple real-robot tasks show consistent improvements over individual policies.

**Strengths:**

- The method provides a unified framework for combining diffusion and flow models trained on different modalities, with a flexibility to support multiple prediction types (noise, score, velocity, etc.).

- The actual algorithm is pretty simple and clear without requiring any additional training.
- Experimental results on Robomimic, PushT, RoboTwin, and four real-robot tasks show consistent gains over each individual policy.

**Weaknesses:**

- My primary concern is the cost of test-time weight search. It involves querying each diffusion/flow policy several times and will likely add considerable inference overhead. How much additional time does this require in simulation and in real-world experiments? I recommend the authors include an analysis regarding this.

- Implementation details for each policy are missing. What is the revised pi0? What are the model architectures and training setups for the diffusion and flow policies? I recommend the authors include these details.

- If I understand correctly, another limitation is that policies being combined must use the same action-chunk length and the same number of diffusion steps. This requirement substantially limits GPC’s applicability to open-source generalist VLAs, which are often trained with different settings.

**Questions:**

- Have the authors tried to combine 3 or more policies? Curious if it can further improve the performance.
- Is there any way to combine different policy classes, such as a flow policy and a diffusion policy with this framework?

I’d be happy to reconsider my score once these points are addressed.

---

> ### Author Response · Authors · 2025-11-18
> **Official Response**
>
> Thank you for the detailed and constructive review! We have included all of your suggestions in the revised version of the paper. Below are our detailed responses:
>
> > **Q1)** Efficiency of GPC
>
> We would like to clarify that we perform weight search only once for each task, by evaluating the success rate at several fixed convex combinations of the parent policies. Once this search concludes, we fix the selected weights and use them for all future rollouts of the composed policy, without any further online adaptation. This one time search cost is manageable. In practice, based on our finding in Sec. 6.3, the search can be optimized to a smaller range ($w\in[0.6,0.9]$) for the stronger one), requiring half the time. This strategy is exactly what we use in our real-world experiments.
>
> We include a comparison of the time cost, showing that GPC is **far more time efficient than fine-tuning or training from scratch.**
>
> | Method | Settings | Time Cost |
> | - | - | - |
> | Train from scratch    | 1M+ demos; N GPUs    |   14d (OpenVLA), 30d (RDT)   |
> | Finetune    | 100 demos; 1 GPU   |   5+ hrs (DP), 8+ hrs (RDT)   |
> | GPC (full search)     | search 9 weights (0.1 to 0.9), each search costs $T_\text{eval}$ = $N_\text{rollout}$ * $T_\text{per-roll}$; $T_\text{eval}^\text{sim}$ = 200 * ~5s = 0.27hr; $T_\text{eval}^\text{real}$ = 20 * ~30s = 0.17hr;|     9 * $T_\text{eval}$, ~ 2.5 hrs
> | GPC (optimized)     | 4 weights (0.6 to 0.9)|   4 * $T_\text{eval}$, ~1 hr |
>
> > **Q2)** Details for policies
>
> We provide details in App. K. We appreciate your feedback and have updated the paper to improve its visibility. To summarize:
> - *Revised Pi-0:* Provided by the robotics community [MimicTest](https://github.com/EDiRobotics/mimictest). It uses `microsoft/Florence-2-base` as the backbone and a linear layer as the action head with flow matching.
> - *General Settings*: All base policies follow their original publicly released code. (a) Train: RoboMimic (1k epochs, bs 1024); PushT (500 epochs, bs 256); RoboTwin: (20k iters, bs 32). (b) Noise Scheduler: RoboMimic & PushT: DDIM (100 train & 10 infer steps); or flow matching (10 train & infer steps). RoboTwin: DP uses DDIM (1k train & 10 infer steps), and RDT with DPMSolver++ (1k train & 5 infer steps).
>
>
> > **Q3)** GPC with different chunk sizes and inference steps
>
> We would like to clarify that GPC is *not* restricted to policies with the same diffusion step count or the same action-chunk length.
>
> (1) Different steps
>
> It is standard practice in diffusion models to use a different number of inference steps at sampling time. Thus, we simply choose an inference time step with a solver to use in GPC. Our experiments follow this protocol and prove the effectiveness.
>
> | Method | Settings | Laptop | Burger | Bowls
> | - | - | - | - | - |
> | DP    | infer step 10    |  0.74    | 0.49 | 0.52
> | RDT    | infer step 5   |  0.69   | 0.46 | 0.47
> | GPC   | infer step 5   |  **0.80**   | **0.57** | **0.66**
>
>
> (2) Different chunk lengths
>
> Handling different action-chunk configs is an interesting setting. Assume chunk lengths differ $H_A \ge H_B$. During sampling, we sample a noise chunk of length $H_A$: policy A predicts a score over the full chunk, while policy B is applied to the first $H_B$ steps. On the overlapping prefix, we apply score composition, and use $\mathbf{s}_A$ for the remaining tail. Here we show it still *improves performance*.
>
> | Method | Settings | Can |
> | - | - | - |
> | DP    | chunk size 8; infer step 5    |    0.50  |
> | FP-D    | chunk size 16; infer step 10   |  0.53   |
> | GPC    | chunk size 16; infer step 10   | **0.68**    |
>
> > **Q4)** Combining three or more policies
>
> GPC can naturally extend to 3 or more policies. Here, we evaluated GPC with 3 base policies, showing that composing 3 policies achieves similar or better results than GPC with 2 policies.
>
> | Method   | Can  | Lift | Square |
> |- |- |-|-|
> | Base Policy (best)     | 0.61 | 1.00 | 0.46
> | GPC (best with 2 policies)  | 0.63 | **1.00** | **0.61** |
> | GPC (DP+MP+FP-D)  | **0.65** | **1.00** | 0.58   |
> | Base Policy (best)  | 0.96 | 0.99 | 0.92  |
> | GPC (best with 2 policies)  | 0.99 | **1.00** | **0.94** |
> | GPC (FP+FP-F+$\pi0$)  | **1.00** | **1.00** | **0.94** |
>
>
> > **Q5)** GPC with flow and diffusion
>
> Conceptually, diffusion models are directly tied to stein score (derivative of log-likelihood w.r.t. data), so adding their scores admits a clear product of probability densities. Flow-matching models instead learn a transport field that is not uniquely associated with a log-density. Due to this representational mismatch, combining a diffusion/score model with a flow-matching model cannot be rigorously interpreted as sampling from the product of their PDFs; such a combination is necessarily heuristic.
>
> ### Conclusion
>
> We appreciate your detailed suggestions and have incorporated all of your feedback into the revised paper. We look forward to further discussion!

---

> ### Author Response · Authors · 2025-11-27
> **Gentle Reminder**
>
> Hi there! Thanks again for the thoughtful feedback. We’ve addressed the questions you raised and incorporated the corresponding updates into the revised version. If there’s anything you feel could benefit from further clarification, we’d be more than happy to continue the discussion.
>
> Best regards,
>
> The Authors

---

### Author Response · Authors · 2025-11-18
**General Response Prior to the Discussion**

We would like to thank all reviewers for the detailed and considerate feedback!

We are happy that all four reviewers agree that GPC is a **novel and effective** framework for multiple policy composition: "Idea of utilizing test-time search is sound" (**vsK6**); "simple and clear" (**BM4P**); "novel and interesting" (**mnxH**); "a novel framework in composing different policies" (**d2UQ**).

We're further glad that reviewers agree that GPC is very **flexible** in different conditions: "flexibility to support multiple prediction types" (**BM4P**); "enables composition across both VA and VLA models" (**vsK6**); "can handle different policies, datasets, and modalities" (**d2UQ**). We also appreciate that the reviewers agree that the method is **theoretical-motivated** and experiments are **comprehensive**.

We agree that additional clarification would make the paper stronger, and are happy to report that we have performed **all** the suggested experiments and incorporated the additional analyses and discussions into the revised version.

### General clarifications
We propose General Policy Composition (GPC), a training-free method that enhances performance by combining the distributional scores of multiple pre-trained policies via a convex combination and test-time search. The method is theoretically motivated and flexible for plug-and-play composition of heterogeneous policies, evaluated by extensive experiments in both simulation and real environments.

- **Extra compute overheads of GPC.** Although GPC needs weight search and queries multiple policies for sampling, it is still **far more time efficient** than fine-tuning or training from scratch, with up to 5x faster than finetuning.
- **GPC can adapt to different diffusion settings.** Reviewers BM4P and d2UQ raised the question of the flexibility of our method across different diffusion configurations. We would like to clarify that **GPC can be applied to policies with different diffusion settings, including different numbers of inference steps and action-chunk sizes.**
- **Selecting optimal weights.** We agree that the weights in GPC are an important factor and treat them as hyperparameters, analogous to the guidance scale in classifier-free guidance: useful default ranges can be established empirically. Our full weight sweeps are therefore **meant to identify** and distill simple practical rules (e.g., biasing weights toward the stronger policy), while providing **concrete usage guidance** and reasonable default settings for future users.


### Additional Results

- **For Reviewer BM4P.** We provided (1) a detailed time analysis of GPC compared with other methods (**Tab. 6**); (2) Experiments to prove GPC can adapt to different diffusion inference steps (**Tab. 8**); (3) different action chunk lengths (**Tab. 8**).
- **For Reviewer vsK6.** We conducted (1) detailed inference time comparisons (**Tab. 7**) and provided potential solutions for achieving faster inference speed; and provided (2) detailed text prompts (**App. K**); (3) In-depth analysis of what reasons contribute to the success of GPC (**App. J**).
- **For Reviewer mnxH.** We added (1) comparisons of inference time (**Tab. 7**), and (2) detailed explanations on weight selection, and the potential outcomes under misaligned conditions.
- **For Reviewer d2UQ.** We supplement (1) detailed inference time comparisons (**Tab. 7**); (2) new results on GPC with multiple policies having different action-chunk sizes (**Tab. 8**).

### Conclusion

We sincerely thank the reviewers for their careful feedback and additional suggestions for evaluation, which we believe will significantly strengthen the paper! We look forward to further discussion and are happy to address any additional questions.

---

### Author Response · Authors · 2025-11-30
**General Response Summary for AC**

Hello AC,

Thank you very much for your time and for overseeing our submission. We are very grateful for your help in coordinating the review process.

Due to the OpenReview incident and the lack of opportunity for additional discussion with the reviewers, we wanted to summarize the reviewers' comments and our efforts to address ***all*** suggestions. We hope this summary will help clarify the contributions of our work, and how our work has been improved by the reviewers' feedback.

### Summary

We are pleased that all four reviewers consistently recognized GPC as a **novel and effective** framework for policy composition: calling it "idea of utilizing test-time search is sound" (**vsK6**), "novel and interesting" (**mnxH**), "simple and clear" (**BM4P**), and "a novel framework in composing different policies" (**d2UQ**).

Reviewers also agreed that GPC is **highly flexible**, noting its "flexibility to support multiple prediction types" (**BM4P**), ability to "enable composition across both VA and VLA models" (**vsK6**), and capacity to "handle different policies, datasets, and modalities" (**d2UQ**). We further appreciate that reviewers highlighted the method's **theoretical** motivation (**vsK6** & **mnxH**) and **comprehensive** experiments.


In addition to the positive feedback, reviewers generally suggested additional clarification regarding the test-time search & inference cost, and more experiments for different diffusion settings. We address **all** of these suggestions and acknowledge that our paper could better present this contribution.

### Updated results and paper revisions

| Clarification Requested  | Reviewers | Our Solution & Revision  |
|--------------|-----------|--------------------------|
| Time analysis of test-time search vs. other methods | `BM4P` | Clarified the full procedure and added benchmarks showing GPC is **significantly more efficient** than fine-tuning. Updated in **Tab. 6**. |
| Inference time comparison | `vsK6`, `mnxH`, `d2UQ` | Demonstrated that GPC’s inference overhead is **modest** (tens of ms per action), still maintaining real-time control. Results added in **Tab. 7**. |
| GPC with different diffusion steps and chunk sizes | `BM4P`, `d2UQ` | Showed GPC works **robustly across diffusion settings** and heterogeneous chunk lengths with consistent gains. Added in **Tab. 8**. |
| 3+ policy composition | `BM4P` | Demonstrated that 3-policy composition performs **as well or better** than 2-policy GPC. Results in **Tab. 9**. |
| Additional clarifications (experiment configs, prompts, weight selection, success reasons) | `BM4P`, `vsK6`, `mnxH` | Expanded details in the revised version: experiment settings (**App. K**), text prompts (**App. K**), weight selection rationale, and analysis of GPC’s effectiveness (**App. J**). |

We are confident that the results in this work would represent a contribution to the machine learning community, and would represent a solid ICLR conference paper after these revisions.

Best,

The authors

---

### Meta-Review · Area_Chair_yuZr · 2026-01-07

**Summary:**

The reviewers generally found the paper's proposal of General Policy Composition (GPC) to be novel, sound, and mathematically well-motivated. They appreciated the framework's flexibility in combining heterogeneous policies (e.g., VA and VLA) without retraining. However, the primary concerns that informed the decision centered on the practical applicability of the method. Specifically, reviewers questioned the computational cost and latency introduced by running multiple policies simultaneously at inference time, as well as the overhead of the proposed test-time weight search. There were also questions regarding the method's robustness, specifically its sensitivity to the convex weights and whether it yields improvements if one parent policy is significantly weaker than the other.

In summary, I believe the authors did a great job during rebuttal to address the concerns raised by the reviewers (see reviewers concerns section for details) and I recommend accepting the paper. Some practicality limitations as acknowledged by the authors also remain which will constrain the overall impact of the paper.

**Reviewer Concerns:**

- Inference Latency (vsK6, BM4P): The authors provided a detailed time analysis (Table 6) demonstrating that the inference overhead is modest (e.g., 0.13s for GPC vs 0.09s for a single policy) and remains within real-time control requirements (20-30Hz). I also believe the argument that composing policies leads to much less overhead than finetuning is valid.
- Cost of Weight Search (BM4P): The authors clarified that the weight search is a one-time process per task/domain, not a per-step calculation, addresssing concerns about operational latency.
- Implementation Details (BM4P): The authors added missing details regarding model architectures and experimental settings to the Appendix.
- Comparison to Baselines (BM4P): New experiments comparing GPC to LoRA fine-tuning were added to demonstrate the trade-offs between inference-time composition and parameter-efficient training.

**Reviewer Scores:**

From my understanding of the reviews and rebuttal, at least 2 out of 3 reviewers would have increased their scores.

---

### Decision · Program_Chairs · 2026-01-26

Accept (Poster)